# Inhibition of mTORC1 by lncRNA H19 via disrupting 4E-BP1/Raptor interaction in pituitary tumours

Ze Rui Wu[1,2], Lichong Yan[3], Yan Ting Liu[1,2], Lei Cao[4], Yu Hang Guo[2], Yong Zhang[1,2], Hong Yao[1], Lin Cai[2], Han Bing Shang[1], Wei Wei Rui[5], Gang Yang[6], Xiao Biao Zhang[7], Hao Tang[1], Yu Wang[8], Jin Yan Huang [9], Yong Xu Wei[1], Wei Guo Zhao[1], Bing Su[3] & Zhe Bao Wu [1,2]

Aberrant expression of long noncoding RNA H19 has been associated with tumour progression, but the underlying molecular tumourigenesis mechanisms remain largely unknown. Here, we report that H19 expression is frequently downregulated in human primary pituitary adenomas and is negatively correlated with tumour progression. Consistently, upregulation of H19 expression inhibits pituitary tumour cell proliferation in vitro and tumour growth in vivo. Importantly, we uncover a function of H19, which controls cell/tumour growth through inhibiting function of mTORC1 but not mTORC2. Mechanistically, we show that H19 could block mTORC1-mediated 4E-BP1 phosphorylation without affecting S6K1 activation. At the molecular level, H19 interacted with 4E-BP1 at the TOS motif and competitively inhibited 4E-BP1 binding to Raptor. Finally, we demonstrate that H19 is more effective than cabergoline treatment in the suppression of pituitary tumours. Together, our study uncovered the role of H19-mTOR-4E-BP1 axis in pituitary tumour growth regulation that may be a potential therapeutic target for human pituitary tumours.

[1] Department of Neurosurgery, Center of Pituitary Tumor, Ruijin Hospital, Shanghai Jiao Tong University School of Medicine, 200025 Shanghai, China. [2] Department of Neurosurgery, First Affiliated Hospital of Wenzhou Medical University, 325000 Wenzhou, China. [3] Shanghai Institute of Immunology, Department of Immunology and Microbiology, Key Laboratory of Cell Differentiation and Apoptosis of Chinese Ministry of Education, Shanghai Jiao Tong University School of Medicine, 200025 Shanghai, China. [4] Department of Neurosurgery, Beijing Tiantan Hospital, Capital Medical University, 100050 Beijing, China. [5] Department of Pathology, Ruijin Hospital, Shanghai Jiao Tong University School of Medicine, 200025 Shanghai, China. [6] Department of Neurosurgery, The First Affiliated Hospital of Chongqing Medical University, 410000 Chongqing, China. [7] Department of Neurosurgery, Zhongshan Hospital, Fudan University, 200032 Shanghai, China. [8] Department of Neurosurgery, Renji Hospital, Shanghai Jiao Tong University School of Medicine, 200127 Shanghai, China. [9] State Key Laboratory of Medical Genomics, Shanghai Institute of Hematology, Ruijin Hospital, Shanghai Jiao Tong University School of Medicine, 200025 Shanghai, China. These authors contributed equally: Ze Rui Wu, Lichong Yan, Yan Ting Liu. Correspondence and requests for materials should be addressed to B.S. (email: bingsu@sjtu.edu.cn) or to Z.B.W. (email: zhebaowu@aliyun.com)

Pituitary adenoma is a common intracranial tumour, accounting for approximately 25% of all intracranial tumours, and approximately 40% of all pituitary adenomas are prolactinomas[1]. Pituitary adenoma clinical syndromes include visual disturbances, infertility and metabolic syndromes due to aberrant hormone production or oncothlipsis[2,3]. Treating these tumours remains a great clinical challenge, especially for drug-resistant prolactinomas and refractory pituitary tumours[1] due to the lack of effective treatment targets and the complicated mechanism of pituitary tumourigenesis. The mammalian target of rapamycin (mTOR) pathway has been reported to be involved in pituitary tumourigenesis and is considered a treatment target; however, the mechanisms by which mTOR affects pituitary tumourigenesis have not been fully elucidated[4–6].

mTOR is an evolutionarily conserved serine/threonine protein kinase that nucleates two structurally and functionally distinct protein complexes, known as mTOR complex 1 (mTORC1) and mTOR complex 2 (mTORC2)[7,8]. mTOR regulates a wide range of cellular processes, including cell growth, proliferation and metabolism, by integrating both extracellular and intracellular cues[9]. mTORC1 contains three core components: mTOR, mLST8 and Raptor. Raptor functions as a scaffold protein to recruit substrates to mTORC1. mTORC1 is mainly involved in the regulation of cellular anabolic processes, such as protein synthesis and lipid synthesis, to promote cell metabolism and cell growth. Dysregulation of mTORC1 has been implicated in a variety of pathophysiological conditions, including cancer[10]. S6K1 and 4E-BP1 are two well-characterized mTORC1 substrates[9]. Phosphorylation of S6K1 by mTORC1 leads to S6K1 activation, which can enhance mRNA translation efficiency by phosphorylating translational regulators such as RPS6, eIF4B and PDCD4[11,12]. Phosphorylation of 4E-BP1 by mTORC1 releases its inhibitory effect on the initiation of cap-dependent translation of certain proteins by promoting the assembly of the eIF4F complex and 5′ cap-dependent mRNA translation[13,14]. Moreover, 4E-BP1 has been shown to directly suppress tumourigenesis[15]. Thus, stringent regulation of 4E-BP1 phosphorylation is important in normal, as well as cancerous cell growth.

Long noncoding RNAs (lncRNAs) are a class of noncoding RNA transcripts that are longer than 200 nucleotides and have biological functions in species from *Drosophila* to mammals[16]. The broad functional capacity of lncRNAs includes roles in chromatin modification, transcriptional regulation and post-transcriptional regulation[16–18]. The lncRNA-H19 gene, encoding the first lncRNA discovered, is located on chromosome 7 in mice and chromosome 11p15.5 in humans[19] and is transcribed from a conserved imprinted gene cluster that also contains the nearby Igf2 gene encoding insulin-like growth factor 2[20]. H19 is a multifunctional lncRNA that regulates embryo development and growth, glucose metabolism, and tumour development[20,21]. There is no previous report of lncRNA H19 regulating the mTOR pathway. The role of H19 in pituitary tumourigenesis is also unclear.

In this study, we aimed to determine the potential role of H19 in pituitary tumour progression. First, we showed that H19 was downregulated in human pituitary tumour tissues, which was associated with poor progression of pituitary tumourigenesis. Furthermore, we revealed that H19 acted as a tumour suppressor, inhibiting pituitary tumour growth by negatively regulating 4E-BP1 phosphorylation. In addition, mechanistic studies demonstrated that H19 bound to and masked the 4E-BP1 TOR signalling (TOS) motif, inhibiting 4E-BP1 recruitment to mTORC1 by disrupting the binding of 4E-BP1 to Raptor.

## Results

### H19 expression is downregulated in human primary pituitary adenomas and is correlated with tumour progression. Previous

studies have demonstrated that lncRNAs play important roles in tumourigenesis in many types of cancer, including breast cancer[22], gastric cancer[23], colorectal cancer[24] and oesophageal squamous cell carcinoma[25], whereas the function of lncRNAs in the initiation and progression of pituitary tumours is still unknown. To identify potential lncRNAs involved in pituitary tumour initiation and development, we performed a lncRNA microarray to profile lncRNA expression in a cohort of normal pituitary glands ($n = 4$) vs. prolactinoma tissues ($n = 5$). We found that 16 lncRNAs were upregulated and 42 were downregulated in prolactinoma tissues compared with those in normal pituitary glands from healthy subjects (Supplementary Table 1). Among these differentially expressed lncRNAs, H19, a maternally imprinted gene[20], was found to be downregulated in all five prolactinoma tissues (Fig. 1a). Moreover, analysis of a previously studied microarray dataset from the Gene Expression Omnibus (GEO) repository database showed that H19 expression was also decreased in gonadotrope tumours compared with that in normal pituitary tissues (GSE26966) (Fig. 1b). To further verify the microarray results, 37 primary pituitary tumour specimens, comprising 9 prolactinomas, 20 nonfunctioning pituitary adenomas, 6 GH (growth hormone) adenomas and 2 ACTH (adreno-cortico-tropic-hormone) tumours, and 3 normal pituitary glands were used (Supplementary Table 2). We found that the H19 expression level was significantly downregulated in the prolactinomas and the other pituitary tumour subtypes compared with that in the normal pituitary glands (Figs. 1c, d). In addition, we found that there was a negative correlation between tumour volume and H19 level (Fig. 1e). These data suggest that H19 downregulation may be associated with the initiation or development of pituitary tumours.

### H19 suppresses tumour cell proliferation both in vitro and in vivo. Both oncogenic and tumour-suppressive properties of H19 have been reported in previous studies[26–31]. Given that H19 expression was downregulated in pituitary tumours and its decreased expression was associated with pituitary tumour progression, we speculated that increasing H19 expression may suppress the growth or initiation of pituitary tumours.

To test this idea, we stably expressed H19 in pituitary tumour GH3 cells (Fig. 2a) and found that H19 overexpression strongly inhibited GH3 cell proliferation (Fig. 2b). H19 overexpression also potently inhibited GH3 cell colony formation (Fig. 2c). To further investigate the function of H19 in pituitary tumour cells, we stably knocked down H19 expression using lentiviral small hairpin RNAs (shRNAs) in GH3 cells (Fig. 2d, Supplementary Fig. 1a). In contrast with H19 overexpression, H19 knockdown in GH3 cells with two independent shRNAs resulted in augmented cell proliferation (Fig. 2e, Supplementary Fig. 1b) and colony formation (Fig. 2f, Supplementary Fig. 1c). To further test whether H19 expression may reduce the viability of human pituitary tumours, primary pituitary tumour cells were infected with H19 expression adenovirus (one prolactinoma, one GH adenoma and two nonfunctioning pituitary tumours). MTS (3-(4,5-dimethylthiazol-2-yl)-5-(3-carboxymethoxyphenyl)-2-(4-sulfophenyl)- 2H-tetrazolium, inner salt) assays determined that H19 could further inhibit the viability of these primary cells (Supplementary Fig. 2a–e).

Notably, the H19 transcript has been reported to harbour microRNA-675 (miR-675), whose overexpression has been shown to suppress cell proliferation in response to cellular stresses and oncogenic signals[27]. To determine if the antitumour effect of H19 was mediated by miR-675, we also overexpressed the miR-675 precursor in H19 knockdown and wild-type GH3 cells but found no anti-proliferation effect as measured by

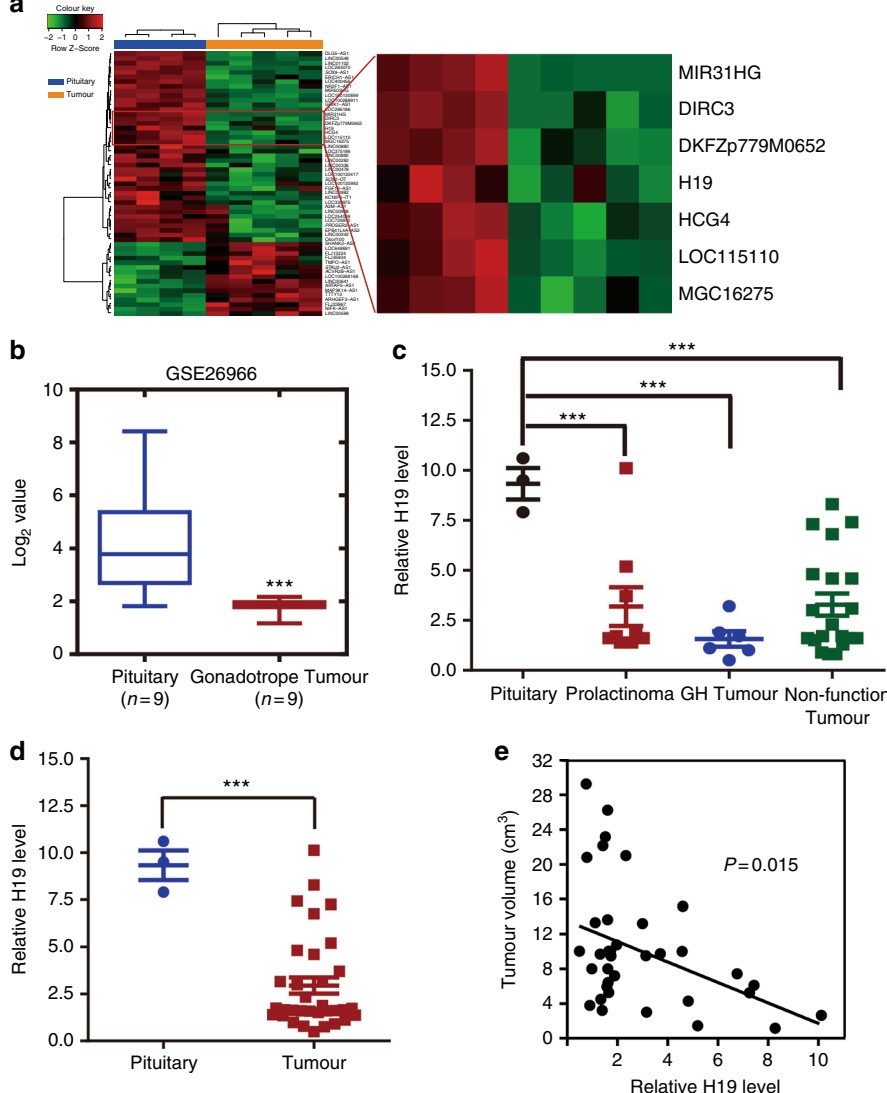

**Fig. 1** H19 expression is decreased in pituitary adenomas. **a** H19 expression is decreased in human prolactinomas compared with that in normal pituitary glands from healthy subjects. The SurePrint G3 Human GE 8 × 60 K Microarray was used to profile lncRNA expression in human normal pituitary glands and prolactinomas (GSE119063). Heatmap showing the differential expression of lncRNAs in normal pituitary ($n = 4$) and prolactinomas ($n = 5$), including the decreased level of H19 in prolactinomas. Right box plot panel shows that H19 expression is decreased approximately 1.51-fold in prolactinomas. **b** H19 expression is decreased in human gonadotropic adenoma (GEO dataset GSE26966). Statistical significance was determined by one-way ANOVA followed by Tukey's multiple comparison test. ***$p < 0.001$. **c** Decreased expression of H19 in prolactinomas, GH tumours and nonfunctioning pituitary tumours compared with that in three normal pituitary tissues. Statistical analysis was determined by paired Student's $t$-test. ***$p < 0.001$. **d** H19 is decreased by 3.24-fold in 37 pituitary tumours compared with that in three normal pituitary tissues. Total RNA was extracted from 37 different subtypes of human pituitary tumours, and qRT-PCR was performed to assess H19 abundance in these pituitary adenomas (PRL = 9, NFPA = 20, GH = 6, ACTH = 2); mRNA levels were normalized to β-actin mRNA. ***$p < 0.001$, two-tailed Student's $t$-test was used for statistical analysis. **e** H19 expression level is negatively corrected with tumour volume in patients ($p = 0.015$, by Pearson's, $n = 37$). Error bars are the mean ± SEM values

proliferation (Supplementary Fig. 3a, b) and colony formation assay (Supplementary Fig. 3c). These data suggest that the anti-proliferation effect of H19 is independent of miR-675 expression.

To evaluate the suppressive function of H19 in pituitary tumourigenesis in vivo, we grafted H19-overexpressing GH3 cells (OE group) and control GH3 cells [empty vector (EV) group] subcutaneously into nude mice. As shown in Figs. 2g–i, H19 overexpression significantly reduced xenograft tumour burden compared with the control group. In contrast to H19 over-expression, knockdown of H19 expression in GH3 cells led to enhance growth of xenograft tumours (Figs. 2j–l, Supplementary Fig. 1d–f, Supplementary Table 3). Of note, the size difference of xenograft tumours in the different

experiments was likely due to the fact that these experiments were done at different time and the tumours collected were grown for different duration. Collectively, these results show that H19 overexpression can inhibit pituitary tumour proliferation in vitro and in vivo.

**H19 suppresses 4E-BP1 phosphorylation in pituitary tumour cells.** To elucidate the underlying molecular mechanisms responsible for the antitumour effect of H19, we assessed the effect of H19 expression on the intracellular signalling pathways that are important for cell proliferation and survival, including the PI3K/AKT/mTOR (phosphatidylinositol 3-kinase/protein

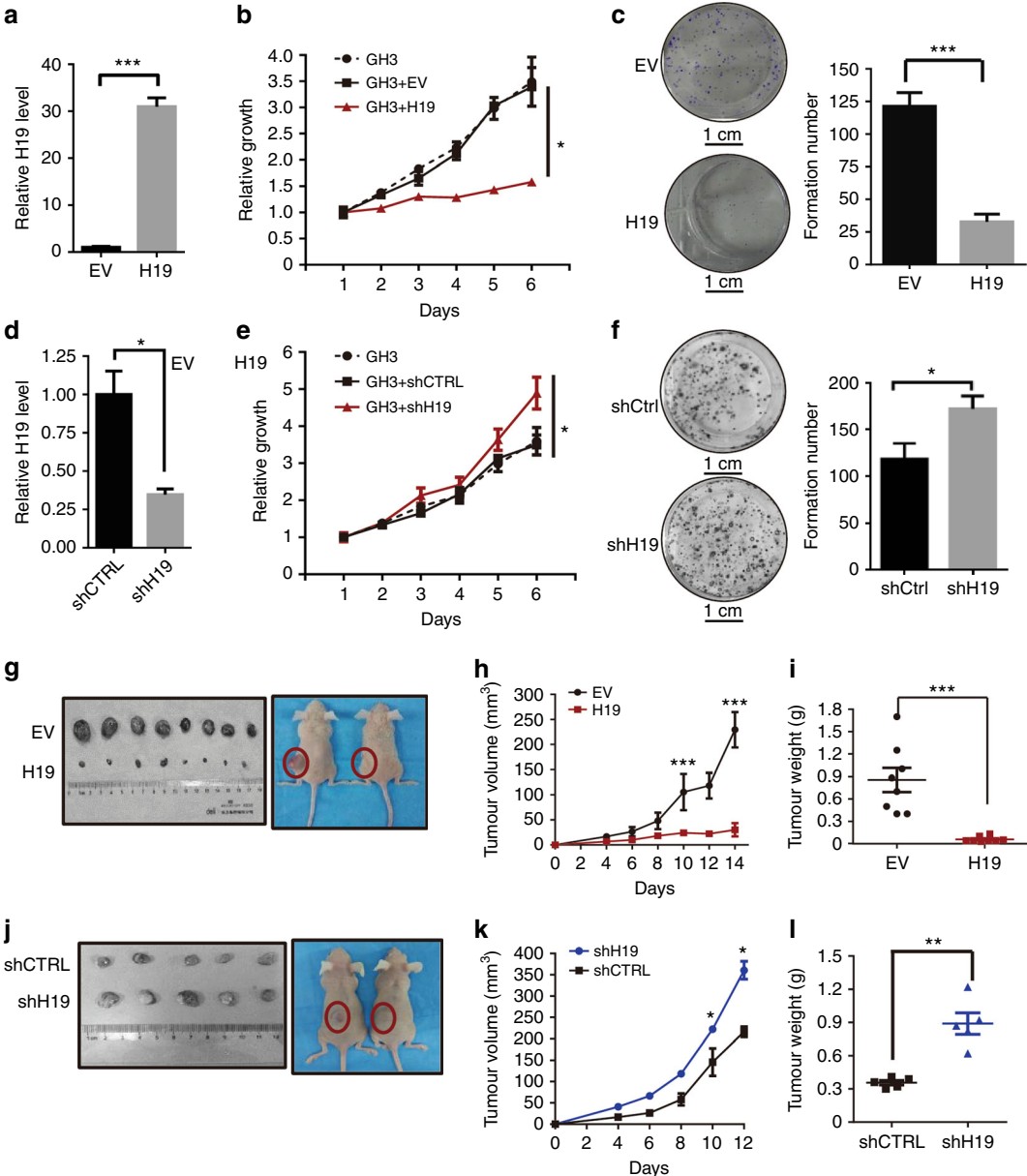

**Fig. 2** H19 inhibits the growth of pituitary tumours both in vitro and in vivo. **a** H19 is stably expressed in GH3 cells with a lentiviral H19 overexpression construct but not in those with a control EV. The H19 level was monitored by qRT-PCR and normalized to GAPDH expression. **b** H19 overexpression inhibits GH3 cell proliferation in vitro. For 6 days, cell viability was periodically measured with an MTS assay ($n = 5$, *$p < 0.05$). **c** H19 overexpression suppresses the colony formation rate of GH3 cells. Cells were seeded into a six-well plate with 200 cells per well and cultured for 10 days, followed by crystal violet staining and colony counting (scale bar,1 cm, ***$p < 0.001$). **d** GH3 cells were infected with lentiviral H19 shRNA or a control shRNA. H19 RNA levels were measured by qRT-PCR and normalized to GAPDH. **e** H19 knockdown enhances GH3 cell proliferation. For 6 days, the proliferation of GH3 cells infected with lentiviral H19 shRNA or a control shRNA was periodically analysed with an MTS assay ($n = 5$, *$p < 0.05$). **f** H19 knockdown enhances the colony formation rate of GH3 cells. Cells stably expressing shH19 or a control shRNA were seeded into six-well plates with 200 cells per well and cultured for 10 days, followed by crystal violet staining and colony counting (scale bar, 1 cm, $n = 5$, *$p < 0.05$). **g** H19 overexpression inhibits GH3 tumour growth in vivo. First, $1 \times 10^6$ H19-overexpressing GH3 cells (OE group) or control GH3 cells (EV group) were subcutaneously grafted into nude mice. At the end of the experiments, mice bearing tumours were sacrificed, and the tumour tissues were collected, photographed (**g**) and weighed (**i**, ***$p < 0.001$). **h** The growth of xenograft tumours was measured by tumour volume every other day (volume = width$^2$ × length × 1/2). **j** H19 knockdown accelerates GH3 tumour growth in vivo. In total, $1 \times 10^6$ H19 stable knockdown GH3 cells (shH19 group) and control GH3 cells (shCTRL group) were subcutaneously grafted into nude mice. At the end of the experiments, mice bearing tumours were sacrificed, and the tumour tissues were collected, photographed (**j**) and weighed (**l**, **$p < 0.01$). **k** The growth of xenograft tumours was measured by tumour volume every other day (volume = width$^2$ × length × 1/2). Error bars are the mean ± SEM values. Two-tailed Student's $t$-test was used for statistical analysis

kinase B/mammalian target of rapamycin)[10], MAPK/ERK (mitogen-activated protein kinase/extracellular regulated protein kinases)[32] and JNK (c-Jun N-terminal kinase)[32] signalling pathways, because their abnormal activities are associated with

tumourigenesis[33]. Surprisingly, H19 expression in GH3 cells had no effect on the basal or inducible activities of ERK1/2, p38, JNK, p65, AKT and the expression levels of RhoA, RhoB, RhoC and Rictor (Supplementary Fig. 4a–c). These data indicate that H19

expression may not affect the PI3K/AKT/mTORC2 pathway, whose activity is dependent on Rictor, and is required for AKT-s473 phosphorylation and Rho protein-mediated cytoskeleton regulation[34,35]. In addition, these data also show that H19 expression does not affect MAPK and NF-κB (nuclear factor κB) pathways. However, we found that H19 overexpression in GH3 and HEK293 cells dramatically suppressed the phosphorylation of 4E-BP1, a well-characterized mTORC1 substrate, but had no effect on S6K1 phosphorylation, another well-characterized mTORC1 target (Fig. 3a, Supplementary Fig. 5). These results suggest that H19 may control mTORC1-mediated 4E-BP1 phosphorylation but not S6K1 phosphorylation. Consistently, knockdown of H19 in GH3 cells strongly increased 4E-BP1 phosphorylation but had little effect on other mTOR targets (Fig. 3b). Furthermore, miR-675 overexpression did not change the 4E-BP1 phosphorylation level (Supplementary Fig. 6a–b). To further test whether H19 expression may reduce p-4E-BP1 levels in human pituitary tumours, primary pituitary tumour cells were infected with H19 expression adenovirus (one prolactinoma, one GH adenoma and two nonfunctioning pituitary tumours). We found that the p-4E-BP1 level was markedly reduced in the H19-overexpressing primary cells (Supplementary Fig. 7a). We also analysed 18 primary pituitary tumours via immunohistochemical (IHC) staining for p-4E-BP1 and divided the p-4E-BP1 scores into two groups: a zero-expression group and a low-expression group (4 vs. 14, Supplementary table 2). Statistical analysis revealed a negative correlation between the p-4E-BP1 score and H19 expression level (Supplementary Fig. 7b). Furthermore, xenograft tumour studies in nude mice showed that the p-4E-BP1 levels were downregulated in vivo upon H19 overexpression and that p-4E-BP1 levels were increased in H19 knockdown tumours; mTORC1-mediated S6K1 phosphorylation, however, was not affected by H19 expression or knockdown (Figs. 3c, d). In addition to inhibiting basal phosphorylation of 4E-BP1, H19 also attenuated mitogen-induced 4E-BP1 phosphorylation without affecting the induction of AKT and S6K1 phosphorylation (Supplementary Fig. 8a, b). Together, these results strongly suggest that H19 can negatively regulate the mTORC1-mediated 4E-BP1 phosphorylation.

4E-BP1 can repress cap-dependent protein translation by competing with eIF4G for eIF4E binding, thus inhibiting the formation of the eIF4F-eIF4G complex at the 5′ cap of mRNA[36]. Phosphorylation of 4E-BP1 by mTORC1 leads to its dissociation from eIF4E, thereby promoting translation of a subset of mRNAs involved in cell survival, proliferation and angiogenesis[13]. Having observed that H19 expression significantly inhibited 4E-BP1 phosphorylation, we speculated that H19 could stimulate the association of 4E-BP1 with eIF4E. To test this hypothesis, we pulled down eIF4E from GH3 cells in the presence or absence of H19 expression using cap analogue m7GTP Sepharose beads. As expected, the association of 4E-BP1 with eIF4E was enhanced upon H19 expression (Fig. 3e). Consistent with a role for 4E-BP1 in translation regulation, we found that H19 overexpression significantly inhibited protein synthesis in GH3 cells as determined by a 35S-Met uptake experiment (Fig. 3f). Taken together, these results indicate that 4E-BP1 might be the key target of H19 in regulating pituitary tumour growth by inhibiting 4E-BP1 phosphorylation and subsequent protein synthesis.

**H19 suppresses pituitary tumour proliferation through the H19–mTORC1–4E-BP1 axis**. Emerging evidence suggests that hyperphosphorylation of 4E-BPs leading to enhanced eIF4E activity is linked to malignant progression in breast, ovarian, prostate and colon cancers[37–39]. Given the specific effect of H19 on 4E-BP1 phosphorylation, we examined whether 4E-BP1 may

directly mediate the H19 effect on the suppression of pituitary tumour cell proliferation and growth. We thus simultaneously knocked down 4E-BP1 and 4E-BP2 expression with two shRNAs in H19-overexpressing GH3 cells (Fig. 4a, Supplementary Fig. 9a) and showed that knockdown of both 4E-BP1 and 4E-BP2 restored the growth and proliferation of H19-overexpressing GH3 cells (Fig. 4b, Supplementary Fig. 9b). Simultaneous knockdown of 4E-BP1 and 4E-BP2 with shRNA lentiviruses also augmented the colony formation of H19-expressing GH3 cells (Fig. 4c, Supplementary Fig. 9c) and promoted the tumourigenic capacity of these cells in nude mice (Figs. 4d–f, Supplementary Table 4, Supplementary Fig. 9d–f). Collectively, these data indicate that H19 suppresses pituitary tumour cell proliferation and tumour growth by blocking 4E-BP1 phosphorylation and function.

**H19 inhibits 4E-BP1 binding to Raptor but not affects mTORC1 complex integrity**. 4E-BP1 and S6K1 proteins are the two most well-characterized direct downstream effectors of mTORC1[40]. H19 expression affected mTORC1-mediated 4E-BP1 phosphorylation with no effect on S6K1 phosphorylation, prompting us to investigate the underlying mechanism of this specificity. The normal S6K1 phosphorylation indicated that the mTORC1 complex should be functional. Indeed, we found that H19 did not affect the mTORC1 complex assembly and integrity (Fig. 5a). Previous studies suggest that Raptor, the core subunit of mTORC1, is responsible for recruiting 4E-BP1 to mTORC1 by recognizing the TOS motif found in its C-terminus[41,42]. This suggests that H19 may impact the binding of 4E-BP1 to Raptor, thus preventing its regulation by mTORC1 without affecting other mTORC1 functions, such as S6K phosphorylation. To test this idea, we performed a co-immunoprecipitation assay to examine the interaction between Raptor and 4E-BP1 in GH3 and HEK293 cells. As shown in Fig. 5b, increased H19 expression attenuated Flag-4E-BP1 binding to HA-Raptor, whereas the interaction between HA-Raptor and Flag-S6K1 was not affected (Fig. 5c). Similar results were also obtained using HEK293 cells (Supplementary Fig. 10a, b). Consistently, H19 expression also affected the endogenous binding of Raptor with 4E-BP1 but not with S6K1 (Figs. 5d, e). These data suggest that the H19-mediated inhibition of 4E-BP1 phosphorylation is due to the inhibition of 4E-BP1 recruitment to mTORC1 by disrupting the Raptor and 4E-BP1 interaction.

**H19 directly binds 4E-BP1 at the TOS domain**. To further understand the mechanism of H19-mediated 4E-BP phosphorylation inhibition, we examined whether H19 could bind directly to 4E-BP1 because we found that H19 resides in the cytoplasm, where it may directly interact with cytoplasmic 4E-BP1 (Supplementary Fig. 10c). Indeed, we clearly detected exogenous H19 RNA in the immunoprecipitation of endogenous 4E-BP1 but not in S6K1, Raptor or AKT1 immunoprecipitations from GH3 cells (Fig. 5f, Supplementary Fig. 10d). In addition, H19 bound to 4E-BP1 in primary pituitary tumour cells (Supplementary Fig. 10e).

Previously, the TOS motif has been reported to be essential for 4E-BP1 to interact with Raptor (Supplementary Fig. 10f). We hypothesized that H19 might suppress 4E-BP1 phosphorylation by masking the TOS motif and thus preventing binding to Raptor[42]. Indeed, full-length 4E-BP1, but not the TOS motif-deleted 4E-BP1 mutant, interacted with H19 RNA (Fig. 5g). Furthermore, we showed that in vitro transcribed H19 could pull-down the full-length 4E-BP1 but not the TOS motif-deleted 4E-BP1 using GH3 whole-cell extracts and purified 4E-BP1 proteins, respectively (Fig. 5h, Supplementary Fig. 10g). To identify the region in H19 that is responsible for 4E-BP1 binding, we constructed a series of H19 deletion mutants and found that the

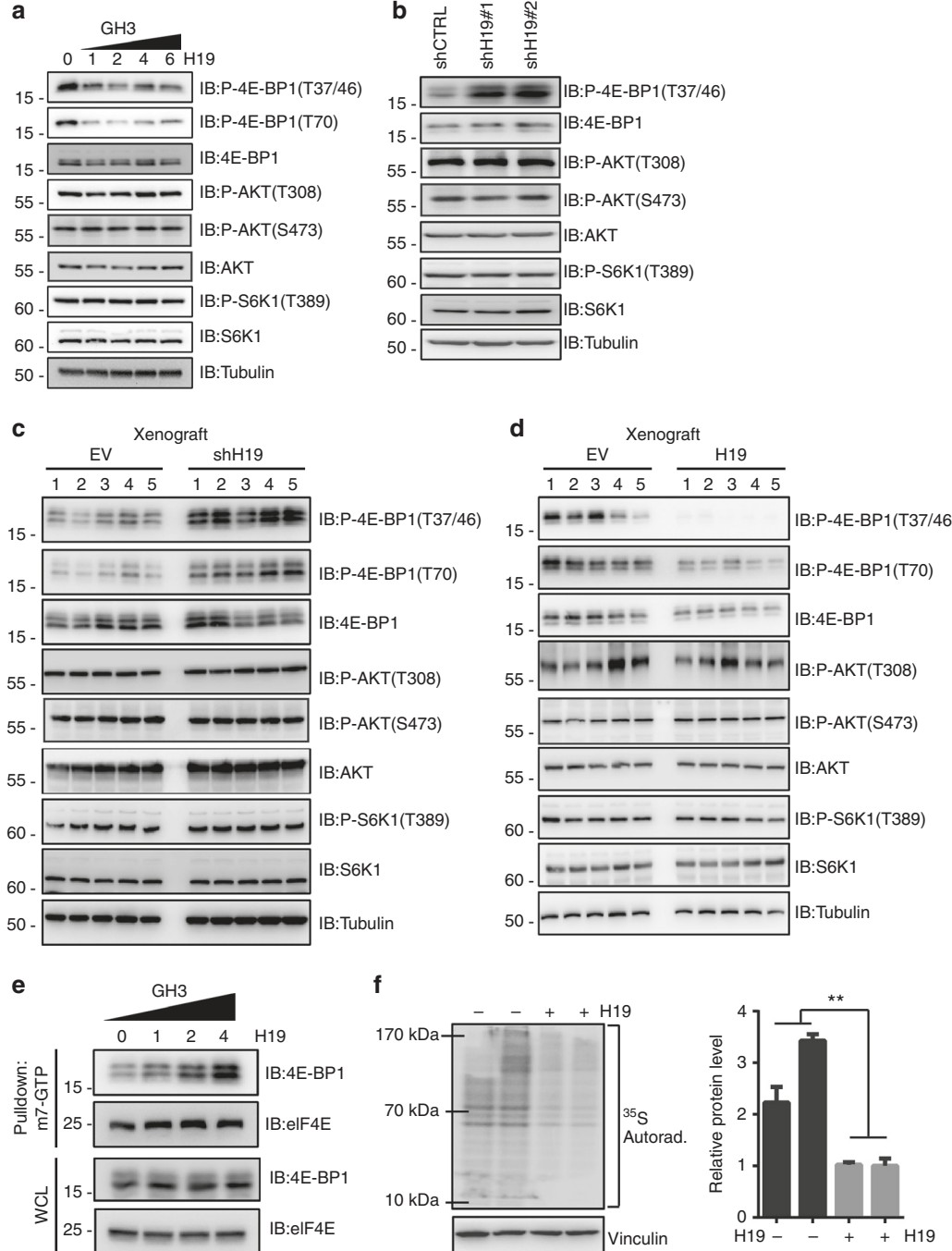

**Fig. 3** H19 decreases the phosphorylation level of 4E-BP1. **a** H19 overexpression suppresses 4E-BP1 phosphorylation in GH3 cells. At 48 h post-transfection, with an increasing dose of H19 in GH3 cells, the resulting cells were collected, and the protein levels of phosphorylated 4E-BP1 Thr70, phosphorylated 4E-BP1 Thr37/46, total 4E-BP1, total AKT, phosphorylated AKT Thr308, total S6K1 and phosphorylated S6K1 were examined by western blot. Tubulin was used as the loading control. **b** H19 knockdown increases the 4E-BP1 phosphorylation in GH3 cells. GH3 cells were infected with lentiviral H19 shRNA1 and shRNA-2 or a control shRNA. The protein levels of phosphorylated 4E-BP1 Thr37/46, total 4E-BP1, total AKT, phosphorylated AKT Thr308, total S6K1 and phosphorylated S6K1 were examined by western blot. **c** Overexpression of H19 suppresses 4E-BP1 phosphorylation in GH3 xenograft tumours. Some of the tumour tissues from Fig. 2g were homogenized in lysis buffer with a tissue homogenizer; then, the samples were centrifuged, and the supernatant was collected and used for immunoblotting analysis with the indicated antibodies. **d** Knockdown of H19 in GH3 xenograft tumours enhances 4E-BP1 phosphorylation. Some of the tumour tissues from Fig. 2j were homogenized in lysis buffer with a tissue homogenizer; then, the samples were centrifuged, and the supernatant was collected and used for immunoblotting analysis with the indicated antibodies. **e** H19 overexpression increases the amount of eIF4E complex bound to the cap analogue m7GTP Sepharose. Whole-cell lysates derived from GH3 cells that were transfected with the indicated dose of H19 were precipitated with m7GTP Sepharose beads followed by immunoblot analysis with the indicated antibodies. **f** H19 inhibits the total translation level in GH3 cells. $^{35}$S labelling of newly synthesized protein was detected by autoradiography (left panel). Quantification of the results using ImageJ (right panel). The experiments were repeated three times. The data shown reflect relative protein synthesis normalized to the control without H19 overexpression. **$p <$ 0.01, two-tailed Student's $t$-test was used for statistical analysis

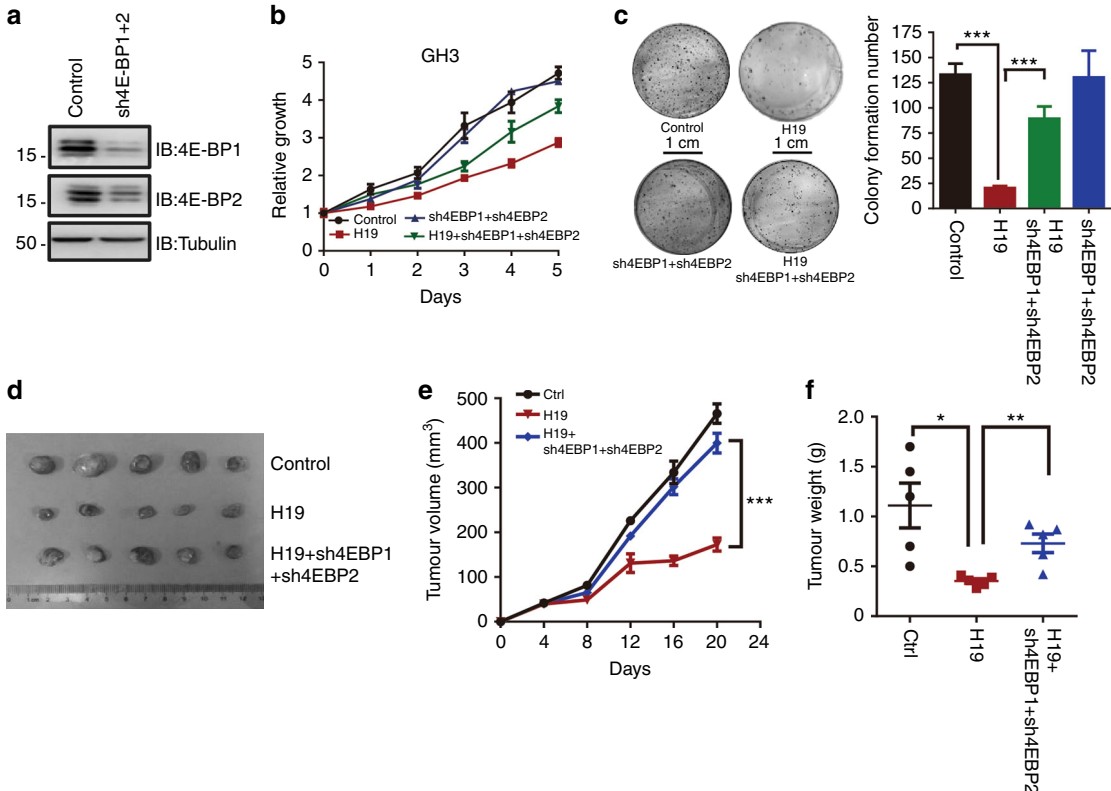

**Fig. 4** H19 suppresses pituitary tumour proliferation by mTORC1–4E-BP1 axis. **a** GH3 cells were simultaneously infected with lentiviral 4E-BP1 shRNA and 4E-BP2 shRNA or a control shRNA. 4E-BP1 and 4E-BP2 protein levels were monitored by western blot. **b** 4E-BP1 and 4E-BP2 double knockdown rescues H19-mediated GH3 cell growth suppression. Cell viability was detected periodically for 5 days using MTS assays and is expressed as relative proliferation (fold change over value on day 1). Error bars represent the SDs in triplicate. **c** A colony formation assay shows that 4E-BP1 and 4E-BP2 double knockdown rescues H19-mediated GH3 cell growth suppression. Cells were seeded into six-well plates with 200 cells per well and cultured for 10 days, followed by crystal violet staining and colony counting (scale bar, 1 cm, ***$p < 0.001$; $n = 3$). **d** 4E-BP1 and 4E-BP2 double knockdown rescues the growth of H19-overexpressing GH3 xenograft tumours. After mice bearing tumours were sacrificed, the tumour tissues were collected, photographed (**d**) and weighed (**f**). **e** The growth of GH3 xenograft tumours was measured by tumour volume. Tumour size was monitored every four days. The data ($n = 5$) were analysed using a sample-paired Student's $t$-test; ***$p < 0.001$. **f** Tumour weight was measured in each group at the end of the experiment. Error bars are the mean ± SEM values. *$p < 0.05$, and **$p < 0.01$

3′ region of H19 is likely involved in 4E-BP1 binding (Fig. 5i). Furthermore, we generated a series of biotinylated H19 fragments (1–650, 651–1300, 1301–1800, 1801–2341) via in vitro transcription, and utilized them in the RNA pull-down assay with GH3 cell lysates. Immunoblot analysis showed that the RNA fragment containing the 3′ sequence between 1801 and 2341 nt of H19 was capable of pulling down 4E-BP1, whereas the other fragments could not (Fig. 5j). Finally, we analyzed the binding of H19 to 4E-BP1 via a cross-linking immunoprecipitation and quantitative PCR (CLIP-qPCR) assay, and the results further narrowed down the major 4E-BP1-binding motif in the 3′ sequence between 2100 and 2200 nt of H19 (Fig. 5k). Together, these data show that the TOS motif of 4E-BP1 is required for its interaction with H19 whereas the 3′ sequence between 2100 and 2200 nt of H19 is responsible for its interaction with 4E-BP1.

Moreover, we transfected wild-type or an 4E-BP1-binding-deficient mutant H19 into H19 knockdown GH3 cells or normal GH3 cells to confirm that the interaction between H19 and 4E-BP1 indeed plays a critical role for pituitary tumour cell proliferation. As expected, overexpression of WT H19, but not the 4E-BP1-binding-deficient H19 mutant, inhibited GH3 cell proliferation (Supplementary Fig. 11a–c). Consistently, the 4E-BP1-binding-deficient H19 mutant lost its ability to suppress 4E-BP1 phosphorylation (Supplementary Fig. 11d–f). Collectively, these data demonstrate that H19 inhibits 4E-BP1

phosphorylation by directly interacting with and masking the TOS motif, thus preventing 4E-BP1 binding to Raptor.

**H19 suppresses oestrogen-induced rat pituitary tumour tumourigenesis.** To further test whether the H19-mediated cell growth inhibition is physiologically important, we constructed a rat prolactinoma pituitary tumour in situ model by administering oestrogen to Fischer 344 rats[43,44]. When tumours were confirmed by magnetic resonance imaging (MRI), the rats were divided into two groups ($n = 4$) that were administered either adenovirus expressing H19 or EVs ($2 \times 10^8$ pfu, via stereotactic pituitary injection). The MRI examination showed that the volumes of tumours treated with H19 adenovirus were significantly less than those of the control rats treated with EV adenovirus 14 days after H19 adenovirus intra-pituitary injection (Fig. 6a, Supplementary Table 5). These data suggest that H19 blocked prolactinoma tumour development. Furthermore, compared with the tumours from the control EV-infected groups, primary tumours from the H19-infected rats showed greatly reduced 4E-BP1 phosphorylation, whereas S6K1 phosphorylation was not affected much (Figs. 6b, c). Consistently, H19-expressing tumours showed markedly reduced proliferation as judged by Ki-67 staining (Fig. 6c). Taken together, these results suggest that H19 effectively inhibits primary prolactinoma

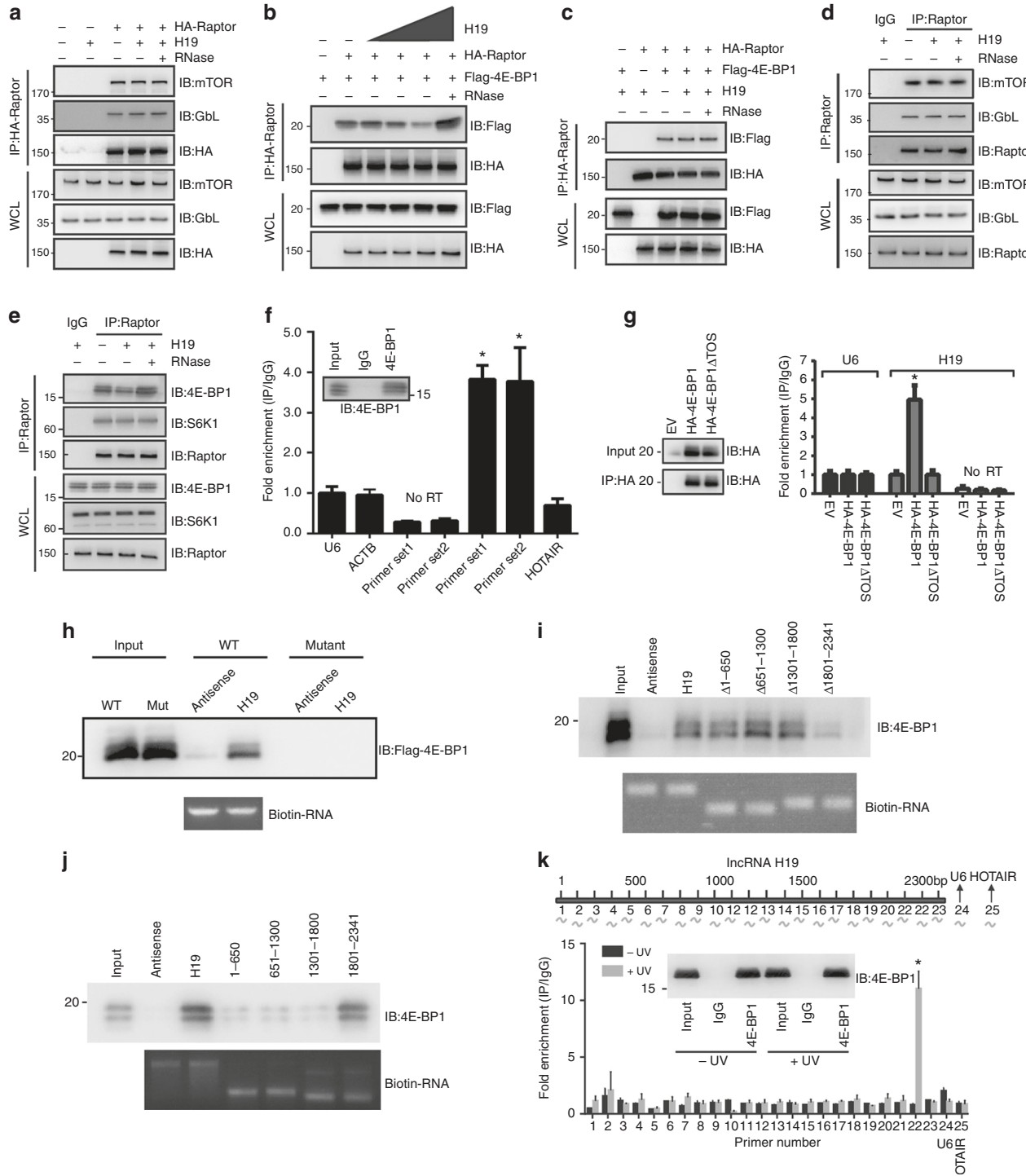

progression by inhibiting 4E-BP1 phosphorylation and that increasing H19 expression may serve as a potential therapeutic strategy for pituitary tumours.

**H19-induced tumour suppression is superior to cabergoline treatment.** Dopamine agonists (DAs), such as cabergoline (CAB), are the first-line treatment for prolactinomas[45,46]. CAB not only inhibits pituitary cell proliferation but also mediates autophagic cell death, as we recently reported[47]. To compare the effect between CAB and H19 overexpression on pituitary tumour growth, we performed xenograft experiments and found that H19 overexpression was more potent in suppressing tumour growth

than CAB (Figs. 7a–c). Interestingly, we found that H19 overexpression led to pronounced inhibition of 4E-BP1 phosphorylation as expected, but CAB treatment had less effect on 4E-BP phosphorylation in the xenograft tumours of GH3 cells (Fig. 7d). Simultaneously, in vitro experiments showed that H19 overexpression significantly suppressed GH3 cell proliferation compared with CAB treatment (Fig. 7e). Interestingly, CAB induced upregulation of H19 expression in a dose- and time-dependent manner (Figs. 7f, g).

Together, these results show that the effect of H19-induced tumour suppression is superior to that of CAB treatment, and thus, H19 may be a potential treatment target.

**Fig. 5** H19 represses 4E-BP1 binding to Raptor by masking the 4E-BP1 TOS motif. **a–c** Immunoblot analysis of whole-cell lysates and immunoprecipitates derived from GH3 cells transfected with the indicated constructs. The RNase treatment group was used as the negative control. **d** H19 overexpression does not affect mTORC1 integrity detected by endogenous co-immunoprecipitation. **e** H19 disrupts the endogenous interaction between 4E-BP1 and Raptor. **f** H19 associates with 4E-BP1 in GH3 cells. Whole-cell lysates of GH3 cells were immunoprecipitated with anti-4E-BP1 antibody or control IgG. The immunoprecipitation was analysed for the presence of H19 via qRT-PCR. Two sets of H19 primers 1 and 2 were used to detect the N- and C-termini, respectively, of full-length H19. Signals were normalized to U6 RNA. Mean ± SD are shown, $n = 3$. *$p < 0.05$. **g** Full-length 4E-BP1, but not TOS domain-deleted 4E-BP1 protein, associates with H19. Anti-HA immunoprecipitation derived from GH3 cells transfected with the indicated HA-4E-BP1 constructs was analysed for the presence of H19 by qRT-PCR. Signals were normalized to U6 RNA. Mean ± SD are shown, $n = 3$. *$p < 0.05$. **h** H19 transcripts and H19 antisense were labelled with biotin and incubated with GH3 whole-cell lysates expression flag-4E-BP1 and flag-4E-BP1 TOS deletion mutant proteins. The capacity of H19 binding to 4E-BP1 proteins was analysed by immunoblotting. **i** The 3′ region of H19 is responsible for 4E-BP1 binding. Full-length and a series of H19 deletion mutants incubated with GH3 whole-cell lysates. The capacity of H19 binding to 4E-BP1 proteins was analysed by immunoblotting. **j** A series of biotinylated H19 fragments (1–650, 651–1300, 1301–1800, 1801–2341) were transcribed in vitro and utilized in the RNA pull-down assay with GH3 cell lysates. The retrieved protein was analysed by immunoblotting. **k** CLIP of 4E-BP1-bound H19 RNA in GH3 cells. RT-qPCR was used to identify the region in H19 bound by 4E-BP1 protein. Location of primer pairs along the H19 RNA indicated in the diagram above. The amount of immunoprecipitated RNAs in each sample is represented as signal relative to the negative (IgG) sample. Mean ± SD are shown, $n = 3$. *$p < 0.05$

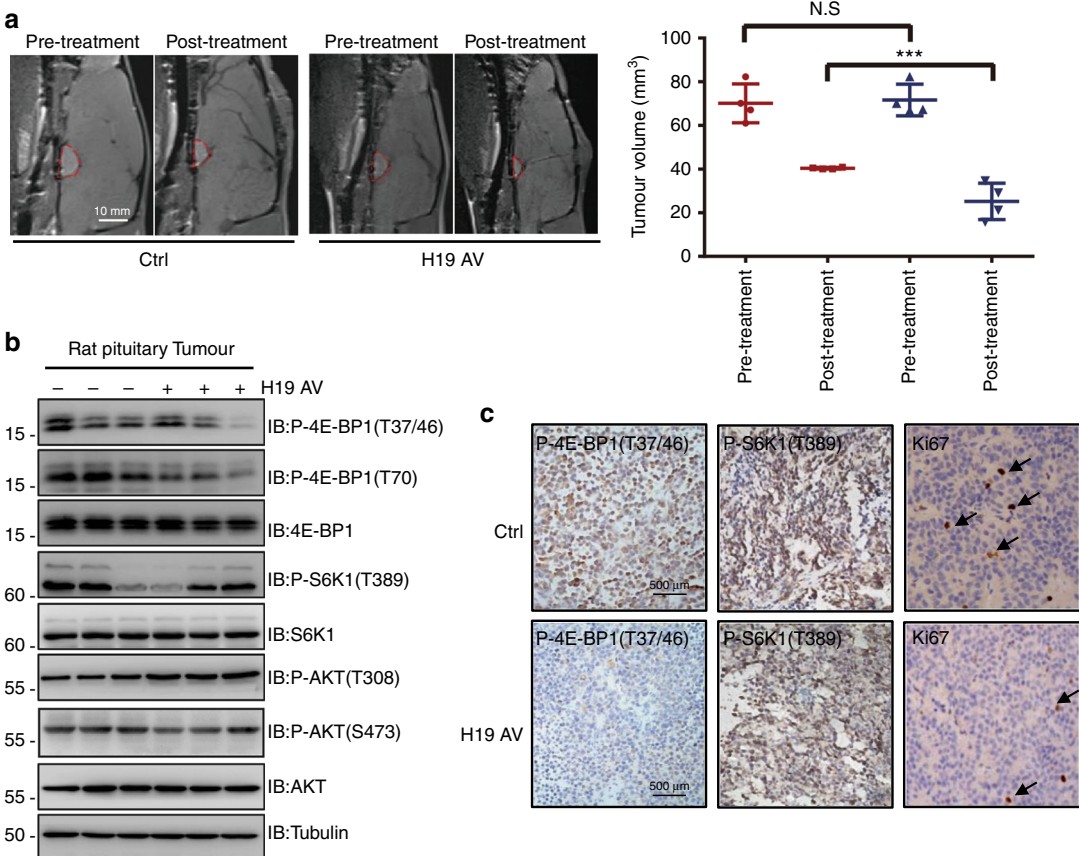

**Fig. 6** H19 suppresses oestrogen-induced rat pituitary tumour tumourigenesis. **a** H19 adenovirus has therapeutic effects on rat in situ prolactinomas. When prolactinomas were induced by 17β-estradiol for 6 weeks, tumour-bearing rats were treated with one stereotactic intratumoural injection of $10^9$ pfu adenovirus harbouring the H19 overexpression construct or the control EV. Two weeks later, tumour size was measured by MRI; tumours were then dissected, and tumour volume was calculated. The data are presented as the mean ± SEM (red circles, $n = 4$; ***$p < 0.001$), scale bar,1 cm. **b** Overexpression of H19 suppresses 4E-BP1 phosphorylation in rat in situ prolactinomas. Some of the dissected tumour tissues described in **a** were homogenized in lysis buffer with a tissue homogenizer; then, the samples were centrifuged, and the supernatant was collected and used for immunoblot analysis with the indicated antibodies. **c** Representative images of IHC staining show that H19 administration decreases 4E-BP1 phosphorylation, but not S6K1 phosphorylation, and pituitary tumour cell proliferation (Ki-67), scale bar, 500 μm

## Discussion

LncRNAs are implicated in the initiation and development of various tumours, but the precise cellular and molecular mechanisms by which they act in these processes remain largely unclear. In this study, we demonstrated that lncRNA H19 acts as a tumour suppressor to inhibit pituitary adenoma growth and progression in vivo and in vitro. Importantly, we unravelled the underlying molecular mechanism of H19-mediated tumour suppression. First, we found that H19 expression was significantly reduced in pituitary tumours. Second, using both in vitro and in vivo models, we demonstrated that overexpression of H19 suppressed tumour cell growth in vitro and tumour growth in vivo, whereas knockdown of H19 expression promoted tumour cell proliferation in vitro and tumour growth in vivo. Third, we

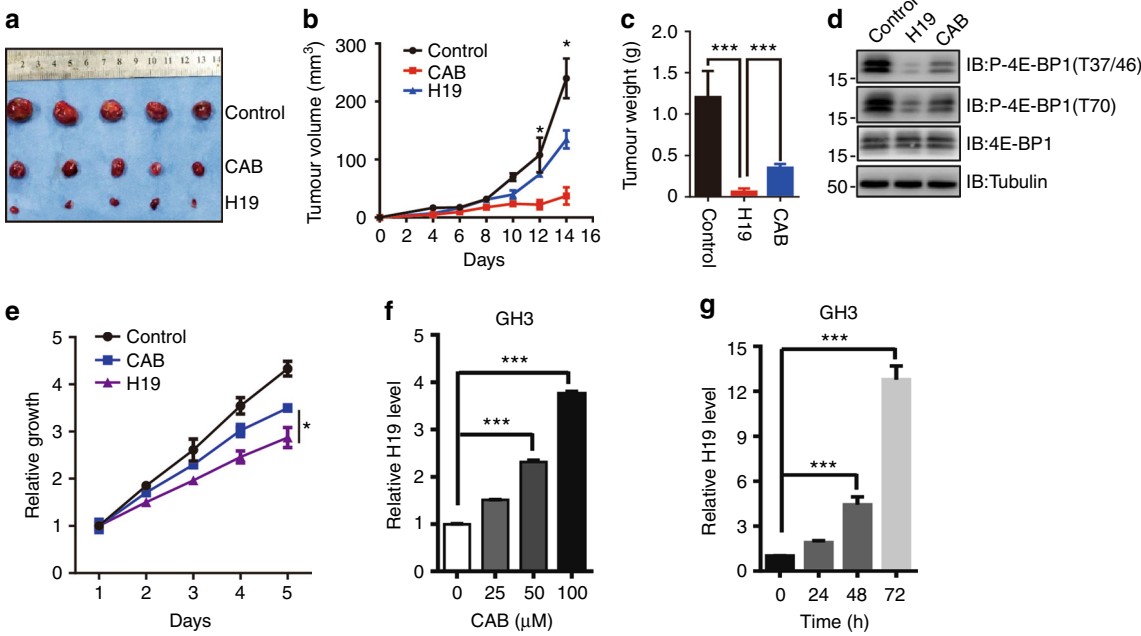

**Fig. 7** H19-induced tumour suppression is superior to CAB treatment. **a** H19 suppresses GH3 tumour growth more potently than CAB treatment (0.75 mg/ kg) in vivo. First, $1 \times 10^6$ GH3 cells with or without H19 stable overexpression were injected subcutaneously into nude mice. The normal GH3 cell mice were divided into two groups that received either gavage with saline solution or cabergoline. **b** The xenograft tumour size was monitored every other day (volume = width$^2$ × length × 1/2). At the end of the experiment, the xenograft tumours were dissected, photographed and weighed (**c**). **d** 4E-BP1 phosphorylation is more intensely suppressed in xenograft tumours treated with H19 than those treated with CAB. Some of the tumour tissues were homogenized in lysis buffer with a tissue homogenizer; then, the samples were centrifuged, and the supernatant was collected and used for immunoblot analysis with antibodies against total 4E-BP1 and phosphorylated 4E-BP1 Thr37/36 and Thr70. Tubulin was used as the loading control. **e** H19 suppresses GH3 cell proliferation more potently than CAB treatment (50 μM) in vitro. **f, g** CAB induces H19 expression in GH3 cells in both a dose-dependent (**f**) and time-dependent (**g**) manner. Total RNA was extracted from GH3 cells, followed by CAB treatment at different CAB doses and periods of time. qRT-PCR was employed to measure the H19 expression level in GH3 cells after CAB treatment. Relative H19 expression was normalized to actin expression. Error bars are the mean ± SEM values, ***$p < 0.001$

showed that H19 could bind to the TOS motif of 4E-BP1 and prevent it from binding to Raptor, leading to selective inhibition of mTORC1-mediated 4E-BP1 phosphorylation without affecting either mTORC2 activity or mTORC1-mediated S6K1 phosphorylation. Our results thus revealed a mechanism by which lncRNA H19 selectively regulates mTORC1 function by inhibiting the interaction of 4E-BP1 and Raptor (Fig. 8).

The mTORC1 effect on cell growth and metabolism can be mediated by multiple downstream effectors, such as S6K1 and 4E-BP1. H19 expression does not impair all the functions of mTORC1; rather, it only reduces 4E-BP1 phosphorylation. Phosphorylation of 4E-BP1 promotes its dissociation from eIF4E, thus allowing it to bind to the mRNA 7-methylguanosine cap structure to initiate the translation process[48]. Numerous previous studies have established the critical role of 4E-BP1-regulated translation initiation for proper control of cell growth, proliferation, differentiation and apoptosis[36,49]. Suppression of the 4E-BP1 dissociation from eIF4E is mainly associated with proliferation defects but not apoptosis in AKT/Ras tumour mice[50]. In this regard, it has been reported that 4E-BP1 could act as tumour suppressor[35,46]. Our data showing that H19 overexpression suppresses 4E-BP1 phosphorylation is consistent with recent clinical findings that high 4E-BP1 phosphorylation levels are associated with poor prognosis in several tumour types, including breast, ovary and prostate tumours[39]. Conversely, inhibition of 4E-BP1 phosphorylation has also been reported to significantly inhibit tumour growth in vivo[51,52]. In our study, H19-mediated inhibition of 4E-BP1 phosphorylation resulted in translation stagnation and pituitary tumour growth arrest in vivo.

Very little is known about the role of lncRNAs in mTOR regulation. Matsumoto et al. found that LINC00961-encoded polypeptide (SPAR) impairs recruitment of mTORC1 to the lysosome and specifically regulates the activation of mTORC1 by amino-acid stimulation[53]. SPAR is localized to the late endosome/lysosome and interacts with lysosomal v-ATPase to negatively regulate mTORC1 activation[53]. However, our study revealed that it is the RNA, rather than its encoded peptides, which specifically determines mTOR substrate specificity.

The mTORC1 substrates S6K1, 4E-BP1 and PRAS40 have been shown to compete for binding to Raptor, the essential component of mTORC1[54]. For instance, PRAS40 negatively regulated mTORC1 activity by directly inhibiting mTORC1 substrate binding, e.g., 4E-BP1 binding to Raptor[55]. In our study, we found that H19 selectively affected the mTORC1-mediated activation of 4E-BP1 but not of S6K1 via inhibiting 4E-BP1 and Raptor binding.

Interestingly, ULK1 was found to significantly inhibit 4E-BP1 phosphorylation, but its target was Raptor, which was phosphorylated at multiple sites by ULK1, resulting in reduced binding of 4E-BP1 by Raptor[14]. GSK3 was also found to inhibit 4E-BP1 phosphorylation by mTORC1; however, the mechanism was also different from our findings, as GSK3 could phosphorylate Raptor at Ser859, which reduced the Raptor–mTOR interaction and mTORC1 complex level, resulting in a reduction in 4E-BP1 phosphorylation[56]. The above regulation of mTORC1 would similarly impact both S6K1 and 4E-BP1.

Although H19 has been well documented as a key regulator of tumour growth and development, early studies suggest that it may act as a tumour suppressor or as an oncogene in different tissues and at different tumour developmental stages[31,57]. For example, H19 was shown to be an oncogene in testicular cancers[58], gastric carcinoma[30], glioma[59] and bladder cancer[60]. In

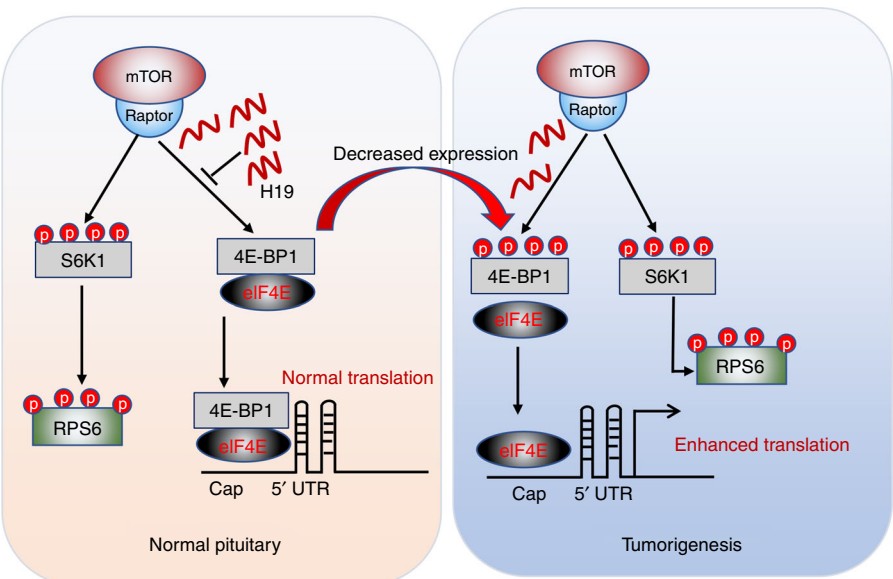

**Fig. 8** A model for H19-mediated regulation of 4E-BP1 phosphorylation. When H19 is abundant (right panel), mTORC1 activity is attenuated because H19 inhibits 4E-BP1 phosphorylation by disrupting the interaction between 4E-BP1 and Raptor. The reduced 4E-BP1 phosphorylation allows 4E-BP1 to bind eIF4E, thus inhibiting its protein translation function. When H19 is absent, 4E-BP1 binds to Raptor and thus is more readily and effectively phosphorylated. This prevents its interaction with eIF4E, resulting in increased protein translation

contrast, H19 functioned as a tumour suppressor in Wilms tumours[61], adrenocortical carcinoma[62], chronic myeloproliferative disorders[26] and Li-Fraumeni syndrome osteosarcomas[63]. Yan et al. reported that H19 serves as a molecular sponge to regulate the bioavailability of the tumour-suppressor miRNA let-7 and promotes tumour cell migration and invasion[29]. Keniry and colleagues found that H19 is a developmental reservoir of miR-675 that suppresses G401 cell proliferation and tumourigenesis[27]. Since our study demonstrates that H19 is a tumour suppressor in pituitary tumours, it would be interesting to know whether H19 acts similarly in other tumours where H19 is a tumour suppressor.

DAs are the first-line treatment for prolactinomas. However, there are resistant prolactinomas and other types of pituitary adenomas that are commonly refractory to DA treatment[1]. GH3 cells, which lack the dopamine 2 receptor[64], the selective target of the DA CAB, are markedly inhibited by H19 (Figs. 2, 6, 7). The H19-mediated tumour growth inhibition appears superior to that of CAB treatment. Although our experimental design has its limitations due to the difficulty in determining the dosages and concentration of CAB to be used, which could affect the outcome of our results, our data show that CAB treatment was less potent than H19 overexpression in suppressing tumour growth in vivo. Our previous studies indicated that CAB inhibits the AKT/mTOR pathway, as well as the reduction in 4E-BP1 phosphorylation[47,65]. Thus, CAB may have a synergistic effect with H19 overexpression in the inhibition of pituitary tumour growth. Finally, 4E-BP1 phosphorylation levels may also be used as an important biomarker for predicting response to therapy. Therefore, the H19–4E-BP1 axis and the CAB–AKT/mTOR axis may be of great therapeutic potential for targeting human pituitary tumours that are refractory to available therapies or surgical treatment.

## Methods

**Plasmids and stable cell lines.** The lncRNA H19 sequence was cloned into the lentiviral vector pLVX-IRES-Puromycin (Clontech Laboratories, Inc.) and pcDNA3.1 for stable and transient expression in mammalian cells. LncRNA H19, the H19 antisense sequence and the various deletion mutants (Δ1–650 bp, Δ651–1300 bp, Δ1301–1800 bp, Δ1801–2341 bp) generated by overlapping PCR were cloned into pGEM-T (Promega) for performing in vitro transcription/

biotinylation reactions. Scramble control shRNA, shRNA targeting different regions of H19 or shRNA targeting 4E-BP1 and 4E-BP2 were cloned into the pLVX-shRNA1 lentiviral vector (Clontech Laboratories, Inc.). The shRNA sequences are listed in Supplementary Table 8. Flag-4E-BP1, Flag-4E-BP1 TOS deletion mutant, HA-4E-BP1, HA-4E-BP1 TOS deletion mutant, HA-Raptor, Flag-S6K1, HA-S6K1 and HA-AKT1 sequences were constructed in the pcDNA3.1 vector. H19, the miR-675 mimic and the miR-675 primer were obtained from RiboBio Co. (Guangzhou, China). Stable GH3 cell lines were generated through transduction with packaged lentivirus. Briefly, HEK293T cells in 10-cm dishes were transfected with pLVX-H19 or pLVX-shRNAs and the packaging plasmids pMD2. G and pSPAX2. Virus-containing medium was collected at 48 and 72 h post-transfection, mixed together, filtered through a 0.45-μm nitrocellulose filter (Millipore), and used to infect GH3 cells in the presence of 5 mg/ml polybrene (Sigma, 107689). Transduced cells were selected for stably infected cells with puromycin (1 μg/ml) in bulk cultures.

**Cell culture.** GH3 cell line was purchased from the American Type Culture Collection (ATCC, CCL-82.1™) without mycoplasma contamination (Tested by Mycoplasma Stain Assay Kit, Beyotime, cat no. C0296, China). GH3 and HEK293 cells were cultured in Dulbecco's modified Eagle's medium (DMEM, Gibco™, 11965–118) supplemented with 10% (v/v) foetal bovine serum (FBS), 100 U/ml penicillin and 100 μg/ml streptomycin (Gibco™, 15140122). Cells requiring stimulation were treated with insulin (100 ng/mL) or EGF (epidermal growth factor, 100 ng/mL) in a time-dependent manner. For primary pituitary tumour cell culture, the pituitary adenoma tissues freshly isolated from surgeries were enzymatically and mechanically dispersed, and the tumour cells obtained were carefully washed by repeated centrifugation and cultured in DMEM with 10% FBS and 1% penicillin/streptomycin (Gibco) in a humidified 5% $CO_2$ cell incubator.

**Cell proliferation assays.** Cells were seeded in 96-well plates at a density of 5000 cells in triplicate. After 24, 48, 72, 96 and 120 h, the number of viable cells was measured with a CellTiter-Glo luminescence cell viability assay (Promega, G3588). Upon addition of MTS solution, the plate was incubated at 37 ℃ for 1–4 h, and the absorbance at 490 nm was determined with a plate reader (TECAN, Switzerland).

**LncRNA microarray.** Total RNA from four pituitary glands and five prolactinomas was subjected to a SurePrint G3 Human GE 8 × 60 K Microarray platform assay performed by BioGenius Co. Ltd, China (GSE119063). The gene expression level was quantified with the software package RSEM. The list of significance was operated by setting the p-value threshold at 0.05. Differential lncRNA expression information is listed in Supplementary Table 1.

**Pituitary tumour samples.** Human pituitary tumour tissue samples were obtained from pituitary tumour patients who underwent surgery between 2015 and 2017 at the Department of Neurosurgery, Ruijin Hospital of Shanghai Jiaotong University (patient information is listed in Supplementary Table 2). Non-neoplastic pituitary

gland specimens were obtained from autopsy. The procedures related to human subjects were approved by the Ethics Committee of Shanghai Jiao Tong University School of Medicine. Written, informed consent was obtained from all patients whose tumour tissues were used in this study.

**Xenograft animals and rat prolactinoma model.** Six-week-old nu/nu athymic female mice were purchased from Shanghai Slack Laboratory Animal Co., Ltd., Shanghai, China (SLAC), and kept under specific pathogen-free (SPF) conditions. One million normal GH3 cells, H19-overexpressing or H19 knockdown GH3 cells were mixed in normal saline and injected subcutaneously. In certain experiments, the mice were treated with CAB at 0.75 mg/kg[47]. Xenograft volumes were evaluated by caliper measurements of two perpendicular diameters and calculated individually using the formula: volume $= a \times b^2/2$ ($a$ represents length and $b$ represents width)[66]. Weights of the mice and tumour dimensions were measured every other day. All the mice were sacrificed, and tumours were harvested, followed by photography and western blotting. All procedures were performed in accordance with the National Institutes of Health Guide for the Care and Use of Laboratory Animals.

For the rat prolactinoma model, rat pituitary tumours were induced by subcutaneously implanting 1-cm Silastic capsules containing 10 mg of 17-β oestradiol in Fischer 344 rats (female, 4 weeks old)[43]. The prolactinomas were induced by 17β-estradiol for 6 weeks, as reported by Cao et al.[44]. Five weeks later, all the prolactinomas were validated via MRI before intra-pituitary injection. The rats were anaesthetised with an intraperitoneal injection of 10% chloraldurate (3.5 ml/kg). Then, 1 μl of the adenovirus vector expressing H19 ($10^{11}$/ml) or the vector control was stereotactically injected into tumours bilaterally. Tumours were injected following coordinates relative to the bregma: 5.4 mm posterior, 9.6 mm ventral and 0.8 mm right and left with the tip of a 26-gauge needle fitted to a 10-μl syringe. Two weeks later, after receiving MRI examinations to measure the tumour size, all the rats were sacrificed, and tumour tissues and sera were collected for further assessments. The volumes of xenograft and rat prolactinomas are listed in Supplementary Tables 3–5.

**Immunoblot analysis and immunoprecipitation.** Cells were normally lysed in cell lysis buffer (50mM Tris, pH 7.5, 120mM NaCl, 0.5% NP40, containing protease and phosphatase inhibitors). For mTOR complex immunoprecipitation, cells were lysed in Chaps cell lysis buffer (40mM Tris, pH 7.5, 120mM NaCl, 0.3% CHAPS, containing protease and phosphatase inhibitors). The total protein concentrations of whole-cell lysates were measured with a Supermax i3 spectrophotometer using Bio-Rad protein assay reagent. The same amounts of whole-cell lysates were subjected to SDS-PAGE and immunoblotted with the indicated antibodies. Images were developed using an LAS4000 system (GE, America). The antibody information is listed in Supplementary Table 6. For immunoprecipitation assays, 1000 μg of whole-cell lysates were incubated with anti-Flag or anti-HA antibody-conjugated agarose beads or antibody (1–2 μg) for 3–4 h at 4 °C, followed by a 1-h incubation with Protein A Sepharose beads (GE Healthcare) if free antibody was used. Immunoprecipitates were washed four times with washing buffer (20 mM Tris [pH 8.0], 150 mM NaCl, 1 mM EDTA and 0.5% NP-40), resolved by SDS-PAGE and immunoblotted with the indicated antibodies. For mTOR complex immunoprecipitation, immunoprecipitates were washed four times with Chaps cell lysis buffer.

**H19 overexpression adenovirus packaging.** H19 was subcloned into a pShuttle vector (Clonetech Laboratories, Mountain View, CA) using EasyFusion Assembly Master Mix (New Cell & Molecular Biotech, Suzhou, China). The resulting H19 expression cassette was then cloned into the E1-deleted region of human adenovirus AdHu5 using I-CeuI and PI-SceI. Following PacI linearization, the recombinant construct was then transferred into HEK293 cells (American Type Culture Collection) for adenovirus packaging. Rescued AdHu5-expressing H19 adenovirus was expended from HEK293 cells and purified by caesium chloride density-gradient centrifugation. Viral particle numbers were measured by spectrophotometry. Adhu5-empty with no insertion in the E1-deleted region was employed as the control virus in this study and generated as described above.

**Colony formation assay.** Cells were seeded in six-well plates (500 cells/well) and grown for 2 weeks. After fixation with 4% paraformaldehyde, fixed cells were stained with 1% crystal violet staining solution (Sangon Biotech, E607309–0100) for 15 min at room temperature. The plates were photographed after extensive washing and air drying.

**RNA extraction and qRT-PCR.** Total RNA was purified from cells using TRIzol reagent (Invitrogen) according to the manufacturer's instructions. RNA concentration was assessed with a NanoDrop 1000 spectrophotometer. Equal amounts of RNA (500 ng) were reverse-transcribed using a cDNA synthesis kit (Takara). Diluted complementary DNAs (cDNAs) (1:5 final) were subjected to qPCR analysis using SYBR Green Supermix reagent (Invitrogen). The quantitative real-time PCR (qRT-PCR) primers are listed in Supplementary Table 7.

**RNA immunoprecipitation (RIP) experiment.** An RIP assay was performed using a Magna RIP RNA Binding Protein Immunoprecipitation Kit (Millipore, MA, USA) according to the manufacturer's instructions. Briefly, whole-cell extracts prepared in lysis buffer containing protease inhibitor cocktail and an RNase inhibitor were incubated at 4 °C for 30 min, followed by centrifugation at $13,000 \times g$ and 4 °C for 15 min. Magnetic beads were preincubated with 5 μg of IP-grade antibody for 30 min at room temperature with gentle rotation. The supernatant was added to the bead–antibody complexes in immunoprecipitation buffer and incubated at 4 °C overnight. Finally, the RNA was purified and quantified by qRT-PCR. Normal rabbit IgG controls were assayed simultaneously to ensure that the signals were detected from RNA that was specifically bound to protein.

**RNA pull-down experiment.** RNA pull-down assays were performed with a Pierce Magnetic RNA-Protein Pull-Down Kit (Thermo Scientific) as described[67]. In brief, the plasmids carrying full-length or different H19 RNA deletion mutants were linearized and transcribed in vitro using RiboMAX Large Scale RNA Production Systems-T7 (Promega). Transcribed RNA was labelled by biotin at the 3′ end using a Pierce RNA 3′ End Desthiobiotinylation Kit (Thermo Scientific). Cells transfected with Flag-4E-BP1 or Flag-4E-BP1 Δ TOS plasmids were collected and lysed in IP lysis buffer (Thermo Scientific) for 30 min at 4 °C with agitation. Pull downs were performed with 10 pmol of biotin-labelled RNA or antisense RNA as the control. The bead–RNA–protein complexes were washed briefly with wash buffer four times. The enriched protein was used to detect the interaction between protein and RNA via an immunoblotting assay.

**CLIP and qPCR (CLIP-qPCR).** The CLIP assay was performed as described[68] with the following modifications. Briefly, rat GH3 pituitary tumour cells were transfected with flag-4E-BP1, and 48 h post-transfection, cells were either cross-linked once at 150 mJ/cm$^2$ or not cross-linked (no-UV control) using 254 nm UV light with a CX-2000 Ultraviolet Crosslinker (UVP) before being lysed. After lysis, cells lysates were treated with RNase T1 (final concentration 500 U/mL, ThermoFisher Scientific) for 6 min then subjected to immunoprecipitation with an anti-Flag or IgG antibody following standard RIP protocol[68]. The immunoprecipitated RNA was purified using acidic phenol (Sangon Biotech, A504194-0100) and ethanol precipitation. After reverse transcription (Takara, RR037A), the resultant cDNA was subjected to quantitative real-time PCR (qPCR) assay. The primers used for qPCR were designed to cover the whole H19 sequence beginning at the 5′ end with a length of 100 bp long and listed in Supplementary Table 7. The amount of immunoprecipitated RNAs in each sample is represented as signal relative to that from the negative (IgG) sample.

**Protein synthesis assay.** Wild-type or H19-overexpressing GH3 cells were washed with phosphate-buffered saline (PBS) and incubated in methionine and cysteine-free DMEM (Sigma, D0422) for 30 min. The $^{35}$S-labelled L-methionine and L-cysteine mix (75 μCi in a 35-mm dish) (PerkinElmer, NEG709A) was then added to the medium, and the cells were incubated for an additional 30 min. The cells were quickly washed with cold PBS and lysed in EBC buffer. Proteins were resolved by SDS-PAGE, and newly synthesized proteins were detected by autoradiography.

**IHC.** Antigens were retrieved from formaldehyde-fixed, paraffin-embedded (FFPE) tumour tissue sections by boiling in sodium citrate buffer (pH 6.0) for 30 min using a microwave histoprocessor. The tissue sections were dehydrated and subjected to peroxidase blocking. IHC staining was performed by incubating tissue sections with primary antibodies overnight at 4 °C, followed by incubation with goat anti-mouse horseradish peroxidase secondary antibody (ab6788, Abcam; 1:200 in 1% BSA/TBST) for 1 h at room temperature. The sections were then exposed to DAB substrate (dissolved in Dako substrate buffer, 760–500, Roche, Indianapolis, IN, USA), followed by Gill's haematoxylin counterstaining and standard dehydration treatment. The staining images were obtained using an Axiovert 200 microscope (Carl Zeiss, Oberkochen, Germany).

**Statistics.** Statistical significance in experiments was assessed using GraphPad Prism software, version 6 (GraphPad Software, La Jolla, CA, USA). Unless otherwise indicated, the data in the figures are presented as the mean ± SEM. Student's unpaired, two-tailed $t$-tests with 95% confidence intervals were used to analyse data involving direct comparison of an experimental group with a control group. The reported $p$-values were adjusted to account for multiple comparisons. For all statistical tests, a 0.05 level of confidence (two-sided) was accepted for statistical significance.

## Data availability

The data that support the findings of this study are available from the corresponding author upon reasonable request. The microarray data have been deposited in the Gene Expression Omnibus (GSE119063).

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

## Acknowledgements

This work was supported by the National Natural Science Foundation of China under grant number 81671371 and 81271523 (Z.B.W.), 31470845 and 81430033 (B.S.), and Shanghai Science and Technology Commission 13JC1404700 (B.S.) and 18XD1403400 (Z.B.W.), and Shanghai Municipal Education Commission-Gaofeng Clinical Medicine Grant Support under grant number 20161407 (Z.B.W.).

## Author contributions

L.C.Y., Z.R.W., T.Y.L., designed and performed the experiments, analyzed the data, prepared the figures and co-wrote the manuscript; Y.H.G. and Y.Z. contributed to the performance of the experiments; H.Y., Y.W. and L.C. contributed to the data analysis. H.T., W.W.R. and B.H.S contributed to study of human pituitary adenoma samples. G.W.Z., G.Y., Y.W., Y.X.W. and X.B.Z. provided the specimens of pituitary tumours. J.Y.H. analyzed microarray data. B.S. and Z.B.W. conceived the idea, designed and supervised the study, analyzed the data and co-wrote the manuscript.

## Additional information

**Competing interests:** The authors declare no competing interests.

