## [Peer Review File · Nature Communications]

Reviewers' Comments:

Reviewer #1:

Remarks to the Author:

In this manuscript, Wu et al. report the regulatory mechanism underlying long non-coding RNA H19 suppress pituitary tumors growth. The authors show that H19 expression is negatively correlated with human primary pituitary tumor progression. Mechanistically the authors showed that H19 could specifically block the mTORC1-mediated phosphorylation of 4E-BP1 through disrupting the 4E-BP1 and Raptor interaction. However, several of the main claims are not supported by supportive data. Of greater concern, many of the experiments presented are poorly designed and data quality is very low. These discrepancies reduce the overall impact of the study in its present form. Overall, the general concept of lncRNA involvement in modulating cancer cell proliferation is not novel. In its current state, with deficiencies in describing unexplored functions of lncRNAs and the lack of data demonstrating the role of H19 in modulating 4E-BP1-Raptor interaction in vivo, the manuscript does not have the quality for publication at Nature Communication.

Major concerns:

1. The authors propose the model that H19 associates with 4E-BP1 and that TOS domain of 4E-BP1 is critical for this interaction in vitro. Although the authors show that H19 interacts with 4E-BP1, which is very superficial, biochemical characterization showing if H19 directly interacts with 4E-BP1 is required to support the proposed mechanism. This includes mapping of the H19 sequence motif responsible for 4E-BP1 binding. It would be important to determine the sequence motif of H19 responsible for 4E-BP1 binding in order to preempt any doubts or concerns regarding the reported findings.
2. Most importantly, can the authors show that H19 specifically binds with 4E-BP1 in actual pituitary tumors as well? So far, the physiological relevance of these findings is unclear. Furthermore, H19 knockout mice are available. The mTORC1-4E-BP1 pathway should be examined in H19 KO mice.
3. The authors' model is that H19 functions essentially as a regulator for the 4E-BP1 and Raptor interaction. A basic test of this model would be to assess the absolute copy number of the H19 versus that of 4E-BP1. lncRNA copy numbers are often low, so are there sufficient copies to make the suggested stoichiometric complexes?
4. The authors suggest that H19 is critical to disrupt the Raptor: 4E-BP1 interaction that regulates the downstream mTORC1 pathway essential for cell proliferation and invasion, but this is never directly demonstrated. This key mechanistic step should be shown by functional rescue experiments to confirm that it indeed happens and in the direct manner postulated. Specifically, rescue experiments using wild-type H19 or the 4E-BP1-binding deficient mutant in H19 knockdown or knockout cells are essential.
5. Given that H19 is a regulator of IGF2 gene expression, I am also surprised that IGF2 expression was not even examined in authors' experimental setting. This also mitigates the validity of the manuscript's conclusion.

Minor concerns:

1. All of knockdown experiments should be performed with two individual shRNAs
2. In this manuscript, the authors used one pituitary tumor cell line to perform overexpression and knockdown experiments sometime. I think the authors at least should compare H19 expression level in a panel of pituitary tumor cell lines, and choose the high level cell line to perform knockdown experiment and choose low level cell line to perform overexpression experiments.

3. Many important controls are missing, which makes it difficult to interpret the results. For all RIP experiments, the no RT control and negative lncRNA control are missing.
4. Why do the authors chose p-AKT (T308) as mTOR kinase substrate in Figs 3a, b.
5. In Figs. 5a-d, the authors need add the panel of RNase treatment as a control.

Reviewer #2:

Remarks to the Author:

This is a very detail manuscript combining molecular biology with xenograft models, which addresses the role of the lncRNA H19. It is a very comprehensive study with interesting findings. However, in my view it does not address the relevance of the findings in the context of human pituitary adenoma (PA). All the major findings are derived from in vitro assays or combination of in vitro and xenografts using GH3 cells. This is an important deficiency of the study, which the authors need to address in the resubmission.

1. The microarray screen used 4 normal human pituitaries and 5 prolactinomas. H19 was found to be downregulated in the prolactinomas. Further analysis, mining published gene expression data, suggests that H19 is also downregulated in gonadotrope tumours.

A major flaw of this analysis is that H19 may not be expressed in lactotrope or gonadotrope cells and therefore, the finding that H19 is not expressed in tumours derived from these cells may not be that surprising.

It is also surprising that the authors do not analyse H19 expression in somatotropinomas, corticotropinomas and NFPA, in particular since a big part of the subsequent study is performed on GH3 cells.

The authors need to elaborate further this initial expression analysis to provide data on the expression of H19 in other pituitary adenomas, and if possible relative to the expression of H19 in the normal hormone-producing cell populations.

2. The reduced expression of H19 in Fig. 1c, are these data representative of all types of PA shown in Suppl. Table 2?? Why do the authors use other tumours in this part of the analysis but not previously? Certainly the authors must have data from the array from all of these samples.

3. Fig. 2g and J should be together using the same scale. This will allow the reader to appreciate the size of the control tumours in both experiments, which do not seem to be the same in the current picture.

4. The model proposed for H19 function is totally based on experiments on GH3 cells and xenografts using GH3 cells transfected with LOF or GOF H19 vectors. This research would gain relevance if the authors could validate their molecular findings in the context of human PA, which are accessible to the authors. The increased levels of p4E-BP1 upon H19 expression are documented in the ms in vitro and by xenografts. This is central for the model depicted in Fig. 8. However, the authors do not validate this critical finding in different types of human PAs (e.g. by western blot). Such analyses would be very relevant to reinforce the conclusions of this study (complementing Fig. 3 d and E). In addition, it could link very well with the expression of H19 in different PA types (requested in Point 1).

5. The IP experiments in Fig. 5 are performed with over-expressed tag-proteins in GH3 cells; do the authors have any data using antibodies against the native proteins and in normal physiological conditions (rather than with over-expressed proteins)?

6. Fig. 6a, could the authors represent the data as dot plots for each individual in the two groups? The authors should include the volumes measured for each mouse in the two MRI sessions (pre- and post-treatment). It is important to show that the volume of the tumours in the mice expressing H19 has decreased post-treatment for all individuals treated, while the control treated in tumour would have remained the same volume or increased. It is important to show the evolution of the tumour in each individual mouse.

7. The experiment described in Fig. 7 seems to be very biased to reinforce the research's conclusions. The normal GH3 cells were transplanted and the mice treated with CB at 50 μ M. Tumour size was compared with mice that were transplanted with GH3 cells stably expressing H19. In my view these experiments are not comparable for many reasons. Pharmacokinetics is not an issue in the H19-over-expressing group but are in the CB treated. Dosage, concentration of CB used, etc could also affect the results. The authors need to emphasize these limitations in the text and expand the description of the experimental design (with dosage, concentration, etc) in M&M.

8. Fig. 2 b and e; what does % of control mean?

9. There are many instances throughout the paper where the English is not ok. The ms needs to be edited.

10. Unless the authors show that the effects of H19 are also observed to other human PA tumours, they should be more specific when referring to the effects of H19 on PA. For example, if all conclusions are based on GH3 xenografts and in vitro studies, it is an overstatement to refer to human PA.

Reviewer #3:

Remarks to the Author:

In this manuscript, Wu et al. investigate the role of long non-coding RNA H19 in pituitary adenomas. By overexpressing or inhibiting lncRNA H19 expression, they showed a negative correlation between H19 expression and cell growth in vitro, as well as tumor growth in vivo. Additionally, the authors found a correlation between H19 expression and 4EBP1-phosphorylation, which is one of the major regulator of translation. They demonstrated that H19 interacts with the TOS motif of 4EBP1, which also mediates the interaction between 4EBP1 and raptor, the core subunit of mTOR complex 1 critical to phosphorylate 4EBP1. The authors proposed that raptor and H19 compete to bind to 4EBP1 via the TOS motif and, consequently, inhibit translation. They also propose the use of H19 as a potential therapeutic target in pituitary tumors.

Overall, this manuscript highlights the importance of lncRNAs in controlling a key hallmark of cancer development downstream of a specific oncogenic pathway underlying pituitary cancer. The study is well designed, provides substantial evidence about the role of H19 in tumorigenesis by employing different in vivo and in vitro models, and proposes the use of H19 as a potential therapeutic target. In principle, this manuscript is suitable for Nature Communications, however the authors should address the concerns listed below before publication.

Major Concerns:

- The rationale of the study is based on the findings that H19 expression is downregulated in prolactinomas compared to normal pituitary glands. The authors performed a lncRNA microarray to detect the differential expression of lncRNAs in normal tissue vs tumor. They chose H19 as their main target; however, based on this analysis, it seems that there are many other lncRNAs which

are also downregulated in tumors. Therefore, the authors should comment on the reasons they specifically focused on H19. More importantly, their validation of H19 expression in pituitary tumors is not impressive. The authors should also compare H19 expression between normal cells and tumors with different grades.

- It was shown that the microRNA-675 is embedded in the H19 RNA. The authors overexpressed miR675 in H19 knocked-down cells to test whether the effects they observed by overexpressing H19 is an indirect effect due to the accumulation of miR675. Based on the results of these experiments, they concluded that miR675 has no effect on cell growth. However, the appropriate experiment should be performed in cells where H19, and likely miR675, is normally expressed, and therefore the exogenous miR675 increases the level of this microRNA above the WT dosage.
- The authors demonstrate that 4EBP1 is dephosphorylated in H19 overexpressed cells and speculate about its consequence in protein synthesis. However, to validate this claim they should perform an assay to test the overall translation rate, such as S35-Met incorporation or polysome trace, in H19 overexpressed cells.
- The authors claim that H19 "directly" interacts with 4EBP1 via TOS motif and it competes with raptor. They do not provide any evidence to show that this interaction is direct, so the reviewer suggests to be more careful about the overstatements; IPs do not provide enough evidence for a direct interaction. Additionally, they should also provide at least one Northern blot to show that when 4EBP1 is immunoprecipitated, it binds to this specific full length lncRNA. Furthermore, although it is characterized, they should also show the effect of TOS deletion on Raptor binding.
- In Figure 6b, it is not clear why only 1 fraction obtained from H19 overexpressed tumor shows low p-4EBP1 levels compared to control ones. The authors should clarify this discrepancy. Why not all the tumors respond to H19 overexpression? In the same context, is there a correlation between decreased level of p-4EBP1 and tumor regression?

Minor Concerns:

- Figure 1A is too small, it is not possible to identify H19.
- In Figure 4e, the tumor volume at Day 14 is different than Figure 2e. Based on the figure legends, these experiments should be similar. How do the authors explain this difference?
- It is not clear why CAB treatment has also an effect on p-4EBP1. Is it also dependent of H19 upregulation in CAB treated cell?
- The title of the Supp Figure 2 legend does not correlate with the figure itself.
- The panels in Supp Figure 3 are not interpretable.

There are several typos in the text: Lanes 193, 312,340, 36

Reviewers' comments:

Reviewer #1, Expertise: non-coding RNA, H19 (Remarks to the Author):

In this manuscript, Wu et al. report the regulatory mechanism underlying long non-coding RNA H19 suppress pituitary tumors growth. The authors show that H19 expression is negatively correlated with human primary pituitary tumor progression. Mechanistically the authors showed that H19 could specifically block the mTORC1-mediated phosphorylation of 4E-BP1 through disrupting the 4E-BP1 and Raptor interaction. However, several of the main claims are not supported by supportive data. Of greater concern, many of the English is not ok. The ms needs to be edited.

ve English speaker.

man PA tumours, they should be more specific wvel. In its current state, with deficiencies in describing unexplored functions of lncRNAs and the lack of data demonstrating the role of H19 in modulating 4E-BP1-Raptor interaction in vivo, the manuscript does not have the quality for publication at Nature Communication.

Major concerns:

1. The authors propose the model that H19 associates with 4E-BP1 and that TOS domain of 4E-BP1 is critical for this interaction in vitro. Although the authors show that H19 interacts with 4E-BP1, which is very superficial, biochemical characterization showing if H19 directly interacts with 4E-BP1 is required to support the proposed mechanism. This includes mapping of the H19 sequence motif responsible for 4E-BP1 binding. It would be important to determine the sequence motif of H19 responsible for 4E-BP1 binding in order to preempt any doubts or concerns regarding the reported findings.

Response: We thank this reviewer for his/her critical and constructive comments. We agree that it is important to map the primary sequence of H19 that is essential for its interaction with 4E-BP1. We performed reciprocal experiments using H19 to pull down 4E-BP1 and showed that H19 could only pull down full-length 4E-BP1 but not the TOS-deleted 4E-BP1 mutant, which strongly suggests that H19 may bind to the TOS region of 4E-BP1 (**Fig. 5h**). In addition, we showed that H19 could bind to purified 4E-BP1 *in vitro* (**Supplementary Fig. 9g**). We tried to identify the region of H19 that is responsible for 4E-BP1 binding by using a series of H19 deletion mutants and found that the 3' region of H19 is mainly involved in 4E-BP1 binding (**Fig. 5i**).

2. Most importantly, can the authors show that 1.H19 specifically binds with 4E-BP1 in actual pituitary

tumors as well? So far, the physiological relevance of these findings is unclear. Furthermore, H19 knockout mice are available. The mTORC1-4E-BP1 pathway should be examined in H19 KO mice.

Response: We agree, and in response to the reviewer's suggestions, we immunoprecipitated endogenous 4E-BP1 and S6K1 with anti-4E-BP1 and anti-S6K1 antibodies from the protein extracts of human primary pituitary tumour cells, and then, we analysed the RNAs that were bound to 4E-BP1 and S6K1 using qRT-PCR. We showed an approximately 4-fold enrichment of H19 in the anti-4E-BP1 immunoprecipitates from human primary pituitary tumor cells despite the observation that the human pituitary tumor generally has decreased H19 expression (**Supplementary Fig. 9e**). The result suggests that H19 is able to bind 4E-BP1 in pituitary tumour cells.

As suggested by this reviewer, we are in the process of obtaining H19^{-/-} mice from Professor Yingqun Huang at Yale University. However, it would take more than 6-8 month to ship, re-derive and establish a colony of this mouse line to China from USA, at the moment, we could not perform the suggested experiments.

So far, the physiological relevance of these findings is unclear.

Response: We respectfully disagree with this reviewer. First, we showed a clear correlation between H19 expression and pituitary adenomas (PA) growth. Second, the majority of previous studies have shown that H19 acts as an oncogene, e.g., by promoting proliferation of glioma¹, gastric cancer², breast cancer³ and neuroblastoma SH-SY5Y cells⁴, but our study in PA found that H19 may act as a tumour suppressor, which is novel and highly interesting. Third, we further reveal the molecular mechanism underlying the action of H19 in pituitary tumour development: specific modulation of the activity of 4E-BP1, one of the most important targets of mTORC1, the master regulator of cell growth. Ample previous studies have shown that 4E-BP1 is critically involved in oncogenic activation pathways to promote tumour growth. Our current study also demonstrated that interfering with 4E-BP1 via H19 markedly inhibited PA GH3 cell growth (**Fig. 4**). Finally, we increased the number of our PA patient samples and further confirmed that reduced H19 expression was well correlated with malignant PA growth (**Fig. 1d-e**). Given that our data show that H19 directly interacts with 4E-BP1 to inhibit its phosphorylation and inactivation by mTORC1 (**Fig. 5f, h**), our data strongly suggest that in these progressive PAs, the augmented tumour growth could be due to reduced H19 expression resulting in insufficient 4E-BP1 inhibition.

3. The authors' model is that H19 functions essentially as a regulator for the 4E-BP1 and Raptor interaction. A basic test of this model would be to assess the absolute copy number of the H19 versus that of 4E-BP1. LncRNA copy numbers are often low, so are there sufficient copies to make the suggested stoichiometric complexes?

Response: We appreciate the reviewer's insightful comment and agree with this reviewer that lncRNA copy numbers are indeed often low in cells. However, although we do not know how many H19 copies are needed to mask all 4E-BP1 proteins in a cell, we do consistently see increased 4E-BP1 phosphorylation when H19 expression is knocked down in GH3 cells via siRNA silencing. In addition, the local H19 concentration in specific cellular compartments may be more critical in dictating the outcome of its impact on the 4E-BP1 and Raptor interaction. Our data clearly show that alterations in H19 expression would impact the 4E-BP1: Raptor interaction, even though it would be difficult to determine *in vivo* what copy number of H19 may be needed. 4E-BP1 has been well documented to inhibit cap-dependent translation by competing with eIF4G for binding to eIF4E. It is possible that H19 only needs to bind to a small portion of the 4E-BP1 protein that is mainly localized in the mRNA 5' cap to prevent its accessibility to mTORC1. In this case, relatively low H19 copy numbers could still have major impacts on its target.

4. The authors suggest that H19 is critical to disrupt the Raptor: 4E-BP1 interaction that regulates the downstream mTORC1 pathway essential for cell proliferation and invasion, but this is never directly demonstrated. This key mechanistic step should be shown by functional rescue experiments to confirm that it indeed happens and in the direct manner postulated. Specifically, rescue experiments using wild-type H19 or the 4E-BP1-binding deficient mutant in H19 knockdown or knockout cells are essential.

Response: Following the reviewer's suggestion, we added more data in the revised manuscript to further support the role of H19 as a key regulator of the 4E-BP1 and Raptor interaction. In particular, we showed that wild-type, full-length H19 but not the 4E-BP1-binding-defective H19 deletion mutant could rescue H19 KO cells to inhibit 4E-BP1 phosphorylation (**Fig. 5h-i** and **Supplementary Fig. 10**).

5. Given that H19 is a regulator of IGF2 gene expression, I am also surprised that IGF2 expression was not even examined in authors' experimental setting. This also mitigates the validity of the manuscript's conclusion.

Response: We thank the reviewer for this thoughtful and critical comment. We analysed IGF2 expression in

human pituitary tumours and found that IGF2 expression was also decreased in pituitary tumour tissues (A). In addition, H19 overexpression or knockdown did not change the expression level of IGF2 (B). Therefore, we conclude that IGF2 did not participate in the H19-mediated regulation process in pituitary tumours.

Minor concerns:

1. All of knockdown experiments should be performed with two individual shRNAs.

Response: Yes, we agree. In response to reviewers' suggestions, we repeated our experiments with additional shRNA in **Fig. 3b** and **Supplementary Fig. 1**. Just state what we have done. In addition, we constructed sh4E-BP1 and sh4E-BP2 for these experiments (**Fig. 4**).

2. In this manuscript, the authors used one pituitary tumor cell line to perform overexpression and knockdown experiments sometime. I think the authors at least should compare H19 expression level in a panel of pituitary tumor cell lines, and choose the high level cell line to perform knockdown experiment and choose low level cell line to perform overexpression experiments.

Response: We thank the reviewer for this thoughtful suggestion. Currently, the most commonly used cell line for pituitary tumour research is GH3. We therefore grew primary tumour cells from human pituitary tumours and infected them with H19 overexpression adenovirus. We show that H19 overexpression inhibited 4E-BP1 phosphorylation in these primary cells; it also significantly affected their proliferation (**Supplementary Fig. 2, Supplementary Fig. 7**).

3. Many important controls are missing, which makes it difficult to interpret the results. For all RIP experiments, the no RT control and negative lncRNA control are missing.

Response: We thank the reviewer for pointing out this neglect. The RT control and negative lncRNA controls are included in the revised manuscript (**Fig. 5**).

4. Why do the authors chose p-AKT (T308) as mTOR kinase substrate in Figs 3a, b.

Response: As the reviewer noted, AKT-(T308) is indeed not a direct substrate for mTORC2. We examined the phosphorylation level of AKT-473, a substrate of mTORC2, in additional sets of experiments. When activated by the phosphorylation of T308, [p-AKT (T308)], AKT acts upstream of mTORC1. We measured p-AKT (T308) to determine if the H19-mediated inhibition of the mTORC1 substrate 4E-BP1 phosphorylation could be due to a change in AKT activation.

5. In Figs. 5a-d, the authors need add the panel of RNase treatment as a control.

Response: We agree and added the RNase treatment panel in the revised **Fig. 5**.

Reviewer #2, Expertise: Pituitary Adenomas (Remarks to the Author):

This is a very detail manuscript combining molecular biology with xenograft models, which addresses the role of the lncRNA H19. It is a very comprehensive study with interesting findings. However, in my view it does not address the relevance of the findings in the context of human pituitary adenoma (PA). All the major findings are derived from in vitro assays or combination of in vitro and xenografts using GH3 cells. This is an important deficiency of the study, which the authors need to address in the resubmission.

1. The microarray screen used 4 normal human pituitaries and 5 prolactinomas. H19 was found to be downregulated in the prolactinomas. Further analysis, mining published gene expression data, suggests that H19 is also downregulated in gonadotrope tumours.

A major flaw of this analysis is that H19 may not be expressed in lactotrope or gonadotrope cells and therefore, the finding that H19 is not expressed in tumours derived from these cells may not be that surprising.

It is also surprising that the authors do not analyse H19 expression in somatotropinomas, corticotropinomas and NFPA, in particular since a big part of the subsequent study is performed on GH3 cells.

The authors need to elaborate further this initial expression analysis to provide data on the expression of H19 in other pituitary adenomas, and if possible relative to the expression of H19 in the normal hormone-producing cell populations.

Response: We thank the reviewer for this critical comment. Our initial findings revealed low H19 expression in prolactinomas, whose incidence accounts for approximately 50% of pituitary tumours⁵. In

addition, Michaelis KA et al. found that H19 expression was also decreased in gonadotrope tumours compared with that in normal pituitary tissues. Therefore, H19 is expressed in prolactinomas and gonadotrope cells and is reduced in tumours from these cells. In response to the reviewer's suggestion, we included more evidence in the revised manuscript regarding H19 expression in somatotropinomas, corticotropinomas and NFPA. We analysed 3 normal pituitary glands and 37 pituitary tumours (PRL=9, NFPA=20, GH=6, ACTH=2). As shown in **Supplementary Table 2** and **Fig. 1d-g**, H19 expression was significantly decreased in prolactinomas as well as in the other subtypes of pituitary adenomas compared with that in normal pituitary glands.

2. The reduced expression of H19 in Fig. 1c, are these data representative of all types of PA shown in Suppl. Table 2?? Why do the authors use other tumours in this part of the analysis but not previously? Certainly the authors must have data from the array from all of these samples.

Response: Yes, the reduced H19 expression is representative of all types of PA. As shown in the above response to specific comment #1, H19 expression was significantly decreased not only in prolactinomas but also in other subtypes of pituitary adenomas (**Fig. 1b-c** and **Supplementary Table 2**).

3. Fig. 2g and J should be together using the same scale. This will allow the reader to appreciate the size of the control tumours in both experiments, which do not seem to be the same in the current picture.

Response: We thank the reviewer for this kind suggestion. As shown, we use the same scale as the reference, although the compression in the diagram is inconsistent, but the scale is the same.

4. The model proposed for H19 function is totally based on experiments on GH3 cells and xenografts using GH3 cells transfected with LOF or GOF H19 vectors. This research would gain relevance if the authors could validate their molecular findings in the context of human PA, which are accessible to the authors. The increased levels of p4E-BP1 upon H19 expression are documented in the ms in vitro and by xenografts. This is central for the model depicted in Fig. 8. However, the authors do not validate this critical finding in different types of human PAs (e.g. by western blot). Such analyses would be very relevant to reinforce the conclusions of this study (complementing Fig. 3 d and E). In addition, it could link very well with the expression of H19 in different PA types (requested in Point 1).

Response: Yes, we agree. In our revised manuscript, we added new data with more experiments, including the expression of H19 in different PA types (**Supplementary Table 2**), qRT-PCR data for H19 expression in different subtypes of human pituitary tumours (**Fig. 1c**), and IHC data of 18 patient samples for p-4E-BP1 levels (**Supplementary Fig. 7b**). We found that the p-4E-BP1 level was low when the H19 level is high in these patient tumour samples (**Supplementary Fig. 7b**). Statistical analysis also revealed a negative correlation between p-4E-BP1 expression and H19 expression (**Supplementary Fig. 7b**).

5. The IP experiments in Fig. 5 are performed with over-expressed tag-proteins in GH3 cells; do the authors have any data using antibodies against the native proteins and in normal physiological conditions (rather than with over-expressed proteins)?

Response: In addition to using the tagged proteins, we further performed IP experiments with antibodies against endogenous 4E-BP1, S6K, and Raptor proteins in GH3 cells, which showed the same results (**Fig. 5d-e**).

6. Fig. 6a, could the authors represent the data as dot plots for each individual in the two groups? The authors should include the volumes measured for each mouse in the two MRI sessions (pre- and post-treatment). It is important to show that the volume of the tumours in the mice expressing H19 has decreased post-treatment for all individuals treated, while the control treated in tumour would have remained the same volume or increased. It is important to show the evolution of the tumour in each individual mouse.

Response: We thank the reviewer for this kind suggestion. We now show our data as dot plots and included volume measurements (revised **Fig. 6a** and **Supplementary Table 5**), but we did not measure the tumour volume during the course of the experiments.

7. The experiment described in Fig. 7 seems to be very biased to reinforce the research's conclusions. The normal GH3 cells were transplanted and the mice treated with CB at 50 μ M. Tumour size was compared with mice that were transplanted with GH3 cells stably expressing H19. In my view these experiments are not comparable for many reasons. Pharmacokinetics is not an issue in the H19-over-expressing group but are in the CB treated. Dosage, concentration of CB used, etc could also affect the results. The authors need to emphasize these limitations in the text and expand the description of the experimental design (with dosage, concentration, etc) in M&M.

Response: We agree with this reviewer's comments about this experiment. In response to the reviewer's suggestion, we now discuss the limitations of the experiments (page 15 line 19-20) and added more detailed descriptions of the experimental design in the Materials and Methods.

Previously, we showed that CAB could effectively treat pituitary tumour GH3 cells at 50 μM *in vitro* and 0.75 mg/kg *in vivo*^{6,7}. This experiment showed that H19 overexpression appeared to be more effective than CAB treatment in arresting tumour growth, suggesting that H19 could be a promising therapeutic target.

8. Fig. 2 b and e; what does % of control mean?

Response: We thank the reviewer for asking this question. The phrase “% of control” means “relative growth” to show the different proliferation capacities of the GH3 cells. We revised our text accordingly.

9. There are many instances throughout the paper where the English is not ok. The ms needs to be edited.

Response: We revised our manuscript with the help of a native English speaker.

10. Unless the authors show that the effects of H19 are also observed to other human PA tumours, they should be more specific when referring to the effects of H19 on PA. For example, if all conclusions are based on GH3 xenografts and *in vitro* studies, it is an overstatement to refer to human PA.

Response: We thank the reviewer for this critical comment. In the revised manuscript, we included new data with human prolactinomas and other subtypes of pituitary adenomas (**Fig. 1c**). We also analysed primary pituitary tumours through IHC staining for p-4E-BP1 (**Supplementary Table 2**) and found that there was a negative correlation between the p-4E-BP1 score and H19 expression level (**Supplementary Fig. 7b**). In addition, H19 could bind to 4E-BP1 and inhibit its phosphorylation in primary pituitary tumour cells (**Supplementary Fig. 7a**). Finally, we added data showing that in the oestrogen-induced rat prolactinoma model, H19 also inhibited tumour volume growth (**Fig. 6**). While we agree with this reviewer that we should be cautious not to overstate the conclusion, our new data suggest that H19 could inhibit the growth of different subtypes of pituitary tumours.

Reviewer #3, Expertise: mTOR signalling (Remarks to the Author):

In this manuscript, Wu et al. investigate the role of long non-coding RNA H19 in pituitary adenomas. By overexpressing or inhibiting lncRNA H19 expression, they showed a negative correlation between H19

expression and cell growth in vitro, as well as tumor growth in vivo. Additionally, the authors found a correlation between H19 expression and 4EBP1-phosphorylation, which is one of the major regulator of translation. They demonstrated that H19 interacts with the TOS motif of 4EBP1, which also mediates the interaction between 4EBP1 and raptor, the core subunit of mTOR complex 1 critical to phosphorylate 4EBP1. The authors proposed that raptor and H19 compete to bind to 4EBP1 via the TOS motif and, consequently, inhibit translation. They also propose the use of H19 as a potential therapeutic target in pituitary tumors. Overall, this manuscript highlights the importance of lncRNAs in controlling a key hallmark of cancer development downstream of a specific oncogenic pathway underlying pituitary cancer. The study is well designed, provides substantial evidence about the role of H19 in tumorigenesis by employing different in vivo and in vitro models, and proposes the use of H19 as a potential therapeutic target. In principle, this manuscript is suitable for Nature Communications, however the authors should address the concerns listed below before publication.

Major Concerns:

1. The rationale of the study is based on the findings that H19 expression is downregulated in prolactinomas compared to normal pituitary glands. The authors performed a lncRNA microarray to detect the differential expression of lncRNAs in normal tissue vs tumor. They chose H19 as their main target; however, based on this analysis, it seems that there are many other lncRNAs which are also downregulated in tumors. Therefore, the authors should comment on the reasons they specifically focused on H19. More importantly, their validation of H19 expression in pituitary tumors is not impressive. The authors should also compare H19 expression between normal cells and tumors with different grades.

Response: We thank this reviewer for the critical comments. As shown in the responses to reviewers 1 and 2 above, we analysed 37 additional primary pituitary tumour specimens with qRT-PCR to determine the expression of H19 in these different pituitary tumour subtypes (**Supplementary Table 2, Fig. 1**). As shown in **Supplementary Table 2** and **Fig. 1d-g**, H19 expression was significantly decreased in prolactinomas as well as in the other subtypes of pituitary adenomas compared with that in normal pituitary glands. Furthermore, we found that H19 expression was negatively correlated with tumour size. Additionally, unlike the majority of previous studies, which showed that H19 acts as an oncogene, e.g., by promoting proliferation of glioma¹, gastric cancer², breast cancer³ and neuroblastoma SH-SY5Y cells⁴, our study in pituitary tumours found that H19 may act as a tumour suppressor, which is novel and highly interesting. Furthermore, we also reveal the molecular mechanism underlying the role of H19 in pituitary tumour

development.

2. It was shown that the microRNA-675 is embedded in the H19 RNA. The authors overexpressed miR675 in H19 knocked-down cells to test whether the effects they observed by overexpressing H19 is an indirect effect due to the accumulation of miR675. Based on the results of these experiments, they concluded that miR675 has no effect on cell growth. However, the appropriate experiment should be performed in cells where H19, and likely miR675, is normally expressed, and therefore the exogenous miR675 increases the level of this microRNA above the WT dosage.

Response: We thank this reviewer for the critical comments. Therefore, we measured the proliferation capacity of normal GH3 cells transfected with the miR-675 mimic and found that this neither affected the proliferation of GH3 cells nor changed the level of 4E-BP1 phosphorylation (**Supplementary Fig. 3, 6**). These results demonstrate that H19 inhibition of pituitary tumour proliferation is independent of miR-675. We revised our manuscript accordingly.

3. The authors demonstrate that 4EBP1 is dephosphorylated in H19 overexpressed cells and speculate about its consequence in protein synthesis. However, to validate this claim they should perform an assay to test the overall translation rate, such as S35-Met incorporation or polysome trace, in H19 overexpressed cells.

Response: We thank this reviewer for suggesting this experiment. We performed the ³⁵S-Met incorporation experiment and found that H19 overexpression indeed significantly inhibited protein translation (**Fig. 3f**).

4. The authors claim that H19 “directly” interacts with 4EBP1 via TOS motif and it competes with raptor. They do not provide any evidence to show that this interaction is direct, so the reviewer suggests to be more careful about the overstatements; IPs do not provide enough evidence for a direct interaction. Additionally, they should also provide at least one Northern blot to show that when 4EBP1 is immunoprecipitated, it binds to this specific full length lncRNA. Furthermore, although it is characterized, they should also show the effect of TOS deletion on Raptor binding.

Response: We thank the reviewer for this critical comment. We agree that our data do not allow us to conclude that the binding is direct. However, northern blotting is not sensitive enough to detect H19 from the 4E-BP1 pull-down assay. Using PCR, we detected both the N- and C-terminal sequences of H19 from the 4E-BP1 pull-down assays, suggesting that full-length H19 was associated with 4E-BP1 (**Fig. 5f**). We

also performed the control experiment as suggested by this reviewer to show that the TOS deletion mutant of 4E-BP1 could not interact with Raptor or bind to H19 (**Supplementary Fig. 9f**). In addition, we used RNA pull-down assays with *in vitro* transcribed H19 RNA and showed that H19 could directly bind to purified 4E-BP1 (**Fig. 5h** and **Supplementary Fig. 9g**).

5. In Figure 6b, it is not clear why only 1 fraction obtained from H19 overexpressed tumor shows low p-4EBP1 levels compared to control ones. The authors should clarify this discrepancy. Why not all the tumors respond to H19 overexpression? In the same context, is there a correlation between decreased level of p-4EBP1 and tumor regression?

Response: The Thr-70 phosphorylation of 4E-BP1 was partially decreased in H19-overexpressing tumours, but phosphorylation of Thr-37 and Thr-46 was significantly decreased in all H19-overexpressing tumours. Our results show that H19 primarily inhibits Thr-37 and Thr-46 phosphorylation but partially inhibits Thr-70 phosphorylation, which is consistent with our model.

We used a protein greyscale analysis to determine the level of p-4E-BP1 (Thr37/46) and found that it was 45.1% lower in the H19-overexpressing group than that in the control group. The tumour volume in the H19-overexpressing group was 37.6% less than that in the control group. These data suggest that there is a possible correlation between a reduced p-4E-BP1 level and tumour regression.

Minor Concerns:

1. Figure 1A is too small, it is not possible to identify H19.

Response: The size of **Fig. 1a** has been increased thanks to the reviewer's suggestion.

2. In Figure 4e, the tumor volume at Day 14 is different than Figure 2e. Based on the figure legends, these experiments should be similar. How do the authors explain this difference?

Response: These could be due to the difference in H19 expression between the two experiments with viral infection. Another possibility is that the slightly different numbers of inoculated GH3 cells between the two experiments affected the results. We believe that these differences would not affect the conclusion of our study.

3. It is not clear why CAB treatment has also an effect on p-4EBP1. Is it also dependent of H19 upregulation in CAB treated cell?

Response: Our previous study showed that CAB could inhibit the AKT-mTORC1 pathway; thus, by itself, it could downregulate p-4E-BP1^{6,7}. In addition, our current study showed that CAB can also upregulate H19 in GH3 cells (**Fig. 7f-g**). Thus, we believe that CAB can inhibit 4E-BP1 phosphorylation in an H19-dependent and H19-independent manner.

4. The title of the Supp Figure 2 legend does not correlate with the figure itself.

Response: We corrected the title and thank the reviewer for the comments.

5. The panels in Supp Figure 3 are not interpretable.

Response: We thank the reviewer for pointing out these errors. We split the figure and make it interpretable now. We corrected them accordingly in our revision.

There are several typos in the text: Lanes 193, 312,340, 36

Response: We thank the reviewer for pointing out these errors. We have corrected accordingly in our revision.

References:

1. Jia P, Cai H, Liu X, Chen J, Ma J, Wang P, *et al.* Long non-coding RNA H19 regulates glioma angiogenesis and the biological behavior of glioma-associated endothelial cells by inhibiting microRNA-29a. *Cancer letters* 2016, **381(2)**: 359-369.
2. Chen JS, Wang YF, Zhang XQ, Lv JM, Li Y, Liu XX, *et al.* H19 serves as a diagnostic biomarker and up-regulation of H19 expression contributes to poor prognosis in patients with gastric cancer.

Neoplasma 2016, **63**(2): 223-230.

3. Zhou W, Ye XL, Xu J, Cao MG, Fang ZY, Li LY, *et al.* The lncRNA H19 mediates breast cancer cell plasticity during EMT and MET plasticity by differentially sponging miR-200b/c and let-7b. *Science signaling* 2017, **10**(483).
4. Zhao L, Zhu Y, Wang D, Chen M, Gao P, Xiao W, *et al.* Morphine induces Beclin 1- and ATG5-dependent autophagy in human neuroblastoma SH-SY5Y cells and in the rat hippocampus. *Autophagy* 2014, **6**(3): 386-394.
5. Asa SL, Ezzat S. The pathogenesis of pituitary tumors. *Annual review of pathology* 2009, **4**: 97-126.
6. Leng ZG, Lin SJ, Wu ZR, Guo YH, Cai L, Shang HB, *et al.* Activation of DRD5 (dopamine receptor D5) inhibits tumor growth by autophagic cell death. *Autophagy* 2017, **13**(8): 1404-1419.
7. Lin SJ, Leng ZG, Guo YH, Cai L, Cai Y, Li N, *et al.* Suppression of mTOR pathway and induction of autophagy-dependent cell death by cabergoline. *Oncotarget* 2015, **6**(36): 39329-39341.

Point-by-point responses to Reviewers' comments:

Reviewers' comments:

Reviewer #1, Expertise: non coding RNA (Remarks to the Author):

Although authors performed certain experiments to address my questions, most of these experiments are irrelevant and failed to answer some of key questions.

1. Authors performed the deletion experiment to map the region of H19 responsible for 4E-BP1 binding. However, this method will not identify the precise sequence motif that I requested to identify. It is hard to believe that Fig. 5i is showing specific interaction of 3' region of H19 with 4E-BP1. As I understand, simply deletion of RNA would change the secondary structure that is important for protein binding. To rule out any potential artifacts here, CLIP (cross-linking immunoprecipitation) should be used in order to analyse 4E-BP1 interactions with H19 or to precisely locate RNA binding motif.

Response:

We agree with this reviewer that H19 (Δ 1801-2341) deletion mutant which showed impaired binding to 4E-BP1 does not necessarily indicate that the 1801–2341 nt sequence of *H19* mediates the interaction with 4E-BP1 protein. It is entirely possible that the impaired interaction is due to altered secondary structure resulting from the deletion. On the other hand, it is equally possible that this sequence is required for H19 interaction with 4E-BP1 although it might not be sufficient. To further address this reviewer's comment, we performed the following pull-down experiments and our data confirm that the 1801–2341 nt sequence of H19 is responsible for its interaction with 4E-BP1:

In order to determine which region in H19 is responsible for its interacting with 4E-BP1 protein, we generated a series of biotinylated H19 fragments (1–650, 651–1300, 1301–1800, 1801–2341) via *in vitro* transcription and utilized them in an RNA pull-down assay with GH3 cell lysates. Our data showed that the RNA fragment containing 1801–2341 nt of H19 was capable of pulling down 4E-BP1 whereas other deletion mutants could not suggesting that the 3' fragment of H19 was directly involved in mediating its interaction with 4E-BP1 (**Figure 5j**). We thank the reviewer's advice for using CLIP (cross-linking immunoprecipitation) to assay for H19 binding motif for 4E-BP1, but unfortunately failed to obtain useful results due to unknown reasons. However, we believe that our new pull-down data (**new data Figure 5h-j**) added to this revision are strong enough for us to draw the conclusion that H19 interacts with 4E-BP1 in a direct manner since similar approach has been

widely utilized to map the interacting region of lcnRNA with its target by others in the field (Ali Zhang *et al.* 2015. **Cell Rep** 13:209-221 Figure 2; Miao-Chih Tsai *et al.* 2010. **Science** 329:689-693 Figure 1; Maninjay K. Atianand *et al.* 2016. **Cell** 165:1672-1685 Figure 6). We do agree that further narrowing down the H19 sequence for 4E-BP1 binding is important for future structural understanding of the mechanism, however, we think that is not the focus or the major point of this study.

2. The authors further use the knockdown or overexpression of H19-induced 4E-BP1 phosphorylation changes to argue that there must be sufficient molecules of H19 and 4E-BP1 proteins to enable a stoichiometric RNA scaffolding mechanism. However, these effects could be indirect. The determination of absolute copy numbers of H19 and 4E-BP1 is required.

Response:

As discussed above, we believe that there is a direct interaction between H19 and 4E-BP1. We agree with this reviewer that determination of the absolute copy number of H19 and 4E-BP1 may help to understand the stoichiometric interaction between H19 and 4E-BP1. However, this would not be sufficient to predict the outcome of the true interaction between these two molecules in cells. We believe that the local concentration rather than the absolute copy numbers is a more important factor to determine the impact of H19 on 4E-BP1. For example, i) the overall expression of H19 could be low but the concentration of H19 in certain sub-cellular compartment could be high enough to exert its inhibitory function on 4E-BP1 phosphorylation; ii) not all the 4E-BP1 proteins in a cell are necessary to be suppressed by H19; it is entirely possible that only those 4E-BP1 proteins resided on mRNA 5' cap are suppressed by H19; iii) unfortunately, there is no reliable way to measure the concentrations of both molecules within a given cellular compartment at this time, thus we believe that even we quantitate the copy numbers of H19 or 4E-BP1, it would not answer the question that this reviewer is asking, never to say that it is still a great experimental challenge to determine the absolute molecule number of a protein in a cell. Therefore, we respectfully disagree with this reviewer that it is necessary to measure the copy numbers of H19 and 4E-BP1 for our current study.

3. The authors suggest that H19 is critical to disrupt the Raptor: 4E-BP1 interaction that regulates the downstream mTORC1 pathway essential for cell proliferation and invasion, but this is never directly

demonstrated. This key mechanistic step should be shown by functional rescue experiments to confirm that it indeed happens and in the direct manner postulated. Specifically, rescue experiments using wild-type H19 or the 4E-BP1-binding deficient mutant in H19 knockdown or knockout cells are essential. To respond this comment, authors simply did overexpression instead of rescue experiments.

Response:

We appreciate these concerns. Based on our results, the 3' region of H19 is largely involved in its interaction with 4E-BP1. Consistently, deletion of the 3' region of H19 resulted in impaired interaction with 4E-BP1. As requested, we performed the experiments using WT and 4E-BP1-binding deficient mutant to rescue H19 knockdown GH3 cells. The new results support our conclusion and are included in **Supplementary Figure 11** in this revision.

4. All of knockdown experiments should be performed with two individual shRNAs. However, most of experiments are remained using only one shRNA. For example, *in vivo* functional experiments in Figs. 2j-l etc.

Response:

We thank the reviewer for the kind suggestion. We designed several new siRNAs to target H19, 4E-BP1 and 4E-BP2. After testing their knockdown efficacy, the most efficient siRNA for H19 and a pair of siRNAs for 4E-BP1 and 4E-BP2 were selected for making shRNA expressing vectors, which were used to generate new GH3 cell lines. New data were shown in **Figure 2j-l, Supplementary Figure 1c-e, Figure 4, and Supplementary Figure 9**.

Overall, I am not sure that the authors have conclusively proved that the mechanism operating here is allostery, although it certainly seems like a good candidate. At this state, this revised manuscript is still not a strong candidate for publication on Nature Communication.

Reviewer #2, Expertise: pituitary adenoma and cancer (Remarks to the Author):

I think the paper has improved significantly by addressing the Reviewers' comments. The authors have addressed these comments very thoroughly.

I have only a minor comment that I would like the authors to include in the final version. The weight of the

xenografts tumours vary remarkably between experiments. For example, in Supplementary Table 3, the shH19 group has a weight that is almost identical to the EV control group. This could suggest that the shCTRL (with scrambled SHRNA) is actually decreasing the tumour weight. As these are different experiments, I can think of many reasons for the variability. However, I don't find any explanation on this. I suggest the authors to include a comment on the section where Suppl. Table 3 is cited to alert the reader on this variability and acknowledge possible reasons for this variability.

In summary, a very nice paper. I congratulate the authors.

Response:

We thank this reviewer for the positive comments. Regarding the two experiments showing the size difference of tumours, we believe that this is most likely due to the fact that the two experiments were done at different time, and the xenografts tumors were also collected for different duration for the tumour growth (14-days in Figure 2h, and 12-days in Figure 2k). As kindly suggested by this reviewer, we added a notion to the revised manuscript to make it more clear (**Page 7 line 19-22**).

Reviewer #3, Expertise: mTOR signalling (Remarks to the Author):

The authors have satisfactorily responded to all my questions and the manuscript is ready for publication.

Reviewer #1 (Remarks to the Author):

Although authors provided additional data regarding H19-4E-BP1 interaction, direct interaction is still required to be demonstrated by CLIP assay, which is essential to convincingly support the core mechanism.

Response:

To address this reviewer's concern, we performed cross-linking immunoprecipitation and qPCR (CLIP-qPCR) assay to further demonstrate a direct interaction between H19 and 4E-BP1. Using this CLIP assay, we narrowed down the interaction motif in H19 to the last 100 nucleotide sequence at the 3' prime. Although not at an individual-nucleotide resolution, we hope that this new result has adequately addressed this reviewer's concern and convincingly demonstrated a direct H19 -4E-BP1 interaction. New data is now added to the revised manuscript as **Fig. 5k**.

With the same reason, absolute copy numbers of H19 and 4E-BP1 must be measured.

Response:

We respectfully disagree with this reviewer that it is necessary to measure the absolute copy numbers of H19 and 4E-BP1 to support for our current study. As we responded in our previous revision, determination of the absolute copy numbers of H19 and 4E-BP1 would not be sufficient to predict the outcome of the true interaction between these two molecules in cells since local concentration rather than the absolute copy numbers is a more important factor to determine the impact of H19 on 4E-BP1. For example, the overall expression of H19 could be low but the concentration of H19 in certain sub-cellular compartment could be high enough to exert its inhibitory function on 4E-BP1 phosphorylation. In addition, not all 4E-BP1 proteins in a cell are necessary to be suppressed by H19 since it is entirely possible that only those 4E-BP1 proteins resided on mRNA 5' cap may be needed to be suppressed by H19.

REVIEWERS' COMMENTS:

Reviewer #1 (Remarks to the Author):

1) First of all, in addition to immunoblot, the full autoradiogram of CLIP assay must be shown. Second, the low and high RNase treatment controls should be included in CLIP assay in order to evaluate whether there has a MW shift due to RNA-protein complex formation. Furthermore, in CLIP assay, the RNase treatment digests RNA to very small pieces. How could it be possible for authors to control RNA digestion size around 100bp using the indicated concentration and incubation time of RNaseT1? If titration was done, the data should be shown since this is such a critical step for this method. What size range of RNA-protein complex did author cut from blot to recover RNAs?

Response: Firstly, we thank this reviewer for his/her detailed questions about our specific CLIP assay. We used a CLIP-qPCR assay to assess the interaction sequence region in H19 with 4E-BP1 following a protocol as reported previously¹. Briefly, we titrated the time points of RNase T1 digesting cell lysates at a suggested concentration (0.5 U RNaseT1/ μ l cell lysate) to select an optimal condition which would digest the RNAs to a size around 100 nt. The results of the titration experiments in two different batches were shown below (**Figure 1** and **Figure 2**). Based on these results, we selected a concentration of 0.5 U RNase T1 per μ L cell lysate for 6 min at 22°C, which could proximately digest the RNAs to around 100 nt.

Figure 1:

Figure 2:

Regarding the request for full autoradiogram of CLIP assay, we guess that the reviewer may ask for a different type of CLIP assay, named HITS-CLIP. As mentioned above, we did not use this method instead we employed an alternative assay called CLIP-qPCR assay, which has been successfully used to map interactions between lnc RNAs and RNA-binding proteins (Adam M Schmitt *et al.* Nature genetics, 2016). Together with several other assays, we believe that we have shown strong evidence to support our hypothesis that 4E-BP1 interacts with H19.

1. Yoon JH, Gorospe M. Cross-Linking Immunoprecipitation and qPCR (CLIP-qPCR) Analysis to Map Interactions Between Long Noncoding RNAs and RNA-Binding Proteins. *Methods in molecular biology* 2016, 1402: 11-17.

2) Regarding the absolute copy number of H19, authors claim that local concentration rather than the absolute copy numbers is more important to determine the interaction between H19 on 4E-BP1. This must be tested, however, and not solely speculated, to be the case. One possible set of experiments would be to examine such interaction by Immuno-RNA FISH. If the authors do not address this point, their model would be totally unconvincing and possibly misleading.

Response: We thank this reviewer for asking this question again but we stand for our previous response that further determination of the absolute H19 copy number would *not change* our major conclusion.